# GPT is becoming a Turing machine:
# Here are some ways to program it

## Abstract

We demonstrate that, through appropriate prompting, GPT-3 can be triggered to perform iterative behaviours necessary to execute (rather than just write or recall) programs that involve loops, including several popular algorithms found in computer science curricula or software developer interviews. We trigger execution and description of **iterations** by **regimenting self-attention** (IRSA) in one (or a combination) of three ways: 1) Using strong repetitive structure in an example of an execution path of a target program for one particular input, 2) Prompting with fragments of execution paths, and 3) Explicitly forbidding (skipping) self-attention to parts of the generated text. On a dynamic program execution, IRSA leads to larger accuracy gains than replacing the model with the much more powerful GPT-4. IRSA has promising applications in education, as the prompts and responses resemble student assignments in data structures and algorithms classes. Our findings hold implications for evaluating LLMs, which typically target the in-context learning: We show that prompts that may not even cover one full task example can trigger algorithmic behaviour, allowing solving problems previously thought of as hard for LLMs, such as logical puzzles. Consequently, prompt design plays an even more critical role in LLM performance than previously recognized.

## 1 Introduction

Large language models (LLMs) [2, 27, 5, 21] are trained on large text datasets, which typically include descriptions of procedures and even computer programs [4]. However, their performance on complex reasoning tasks remains limited despite using advanced prompting methods, such as Chain-of-Thought (CoT) [31, 40, 20, 38, 37, 42, 6, 36, 15, 11, 12]. This implies that despite the massive number of parameters and self-attention to all previous tokens, current LLMs are unlikely to solve problems that require many (or iterated) reasoning steps in a direct, savant-like manner. New benchmarks target these more complex tasks, such as logical deduction and logical grid puzzles in BIG-bench Lite [32], and in-context learning of these problems is typically poor. Practical applications like GitHub Copilot show a mix of promise and limitations: Copilot can auto-generate substantial amounts of code [25, 4], but falls short of expert programmers, lacking execution, state tracking, and debugging abilities (apart from anecdotal evidence, e.g. Fig. 3.7 in [3]; see Section A.3).

LLMs generate tokens in order, each based on many previous tokens in the sequence, whether these tokens were part of the prompt or had just been generated by the LLM itself. Such self-attention could allow an LLM to use all previously generated tokens as the scratchpad for tracking reasoning steps, states, etc.[1]. Such use of generated tokens would resemble a classical Turing Machine with its memory tape [34]. In principle, a non-trivial recurrent transformer model with infinite attention could

---

[1]This is likely to be one of the reasons for the increased performance of CoT prompting.

Submitted to 37th Conference on Neural Information Processing Systems (NeurIPS 2023). Do not distribute.

be Turing-complete and capable of executing arbitrary routines, as long as the attention mechanism can be controlled stringently enough. But, even in relatively simple settings, LLMs appear to resist strict controls, e.g., slight changes in prompts can yield dramatically different responses [14, 17, 30], because many recurrent patterns in the training data are encoded into a single model, and learned patterns overlap and vary in the context size. Thus it is easy to mislead with a prompt with accidental alphabetical or numerical ordering, or some undetectable semantic bias [41, 16, 19].

In Section 2, we introduce much stricter attention controls that instruct LLMs to unroll reasoning steps of a procedure with the initially undetermined length, and decide when the solution is found: **Iteration by Regimenting Self-Attention (IRSA)**. The basic way to achieve such deliberate self-attention control is through highly structured prompting with an example of execution path for one example, as illustrated for Bubble Sort algorithm in Prompt 1, which encourages an LLM to output not just the sorted sequence but also the swap count (response in Prompt A.1 in Appendix), which is a challenging task to solve in a savant manner. We further explore **fragmented prompting** which combines multiple fragments of execution paths, as well as the strategy of skipping parts of generated text when performing self-attention. We also discuss interpreter/compiler prompts that can translate an algorithm in a high-level programming language into an IRSA prompt that GPT-3 can execute (with details in Section 2.4 in the Appendix).

We present results on a wide range of algorithms taught in computer science curricula and used to test software engineers in coding interviews, including string manipulations, dynamic programming, and stack operations in Section 3. Our findings point to broader applications for LLMs beyond existing uses like Copilot in areas like software engineering and education [7, 24, 28, 18]. More pressingly, they point out a critical issue in evaluating in-context learning of LLMs, suggesting that current evaluations may underestimate LLMs' abilities if prompts can combine natural language instructions with algorithmic iterative reasoning. The sensitivity of the performance to prompt design may be amplified by the iterative reasoning triggered by the prompt, which will then beg the question: If one LLM beats the other on a task, is it simply because we have not found the right prompt for the second model? For example, IRSA prompting increases the performance of GPT-3 family on logical deduction puzzles from 32% to 76% (Table 1. The discussion in the Appendix also includes an experiment with GPT-4 [21] on a well-known dynamic programming task showing that even the latest member in the family cannot consistently execute code without prompting in IRSA style.

# 2 Iteration by Regimenting Self Attention (IRSA)

Prompt 1, as well as the prompts 2, A.4, A.5, and A.6 in the Appendix, illustrate the basic IRSA. In each of these examples, a single prompt is provided for a task, which, when combined with a new instance of the task, trigger the execution of an iterative algorithm. The algorithms are single loop (Prompts A.5 and A.6) or double loop (Prompts 1, A.4, and 2); and may have a known or unknown number of iterations until termination.

Crucially, the prompts show all state changes and *explain each change before it occurs*. Although the explanation is colloquial, the structure of it is both rigid and repetitive, strictly regimenting the attention to the rules (corresponding to program instructions) and state changes. In all these examples, this strategy hardens the attention sufficiently to facilitate disciplined procedural reasoning, while leaving non-regimented content open to interpretation. For example, Prompt 1 shows how a sequence of 4 integers can be sorted in some detail, but the same prompt can also be used to sort characters alphabetically or animals by size, and the procedure typically works for both shorter and longer lists.

These prompts could be thought of as an instance of Chain-of-Thought prompting [38, 37, 42, 6, 36, 15, 11]. However, a significant distinction lies in the number of reasoning steps, which is limited and fixed in usual CoT applications. In contrast, the algorithms explored here require an unspecified number of iterations required to complete the execution, and may even incorporate double loops. The prompt contains the condition for declaring the end of execution.

## 2.1 Using IRSA to reason over logical puzzles.

Iterative reasoning is required in solving a number of NLP word prolems, (e.g., [32]), not just in execution of standard algorithms, as these algorithms can be employed **after parsing the world problem**. The BIG-bench Logical Deduction task asks for the ordering of several objects given their pairwise relationships described in natural language (e.g., a robin is standing on a branch to

**Prompt 1.** Bubble Sort: The prompt describes iterative state evolution, including counting swaps, and making the determination when to stop. Playground link (use with 0 temperature)

```
Problem: 2, 3, 1, 5
EXECUTION
    Prep
    Length of the list: 4
    Number of consecutive pairs: 3
    a=[2 3 1 5]
    set n_swaps=0
    EndPrep
    Iteration:
        set swap_flag=false. The state is:
        State: a=[2 3 1 5], n_swaps=0, swap_flag=false EndState
        Pair a[1,2] = [2 3] Check if 2<3. Is it true? Yes.
                            Because of that, we leave state as is
        State: a=[2 3 1 5], n_swaps=0, swap_flag=false
        Pair a[2,3]= [3  1] Check if 3<1.  Is it true? No.
                            Thus, we swap_flag=true, increase n_swaps by one,
                            and in the latest a=[2 3 1 5]
                            swap 3 and 1 to get into state:
        State: a=[2 1 3 5], n_swaps=1, swap_flag=true EndState
        Pair a[3,4]= [3 5]  Check if 3<5. Is it true? Yes.
                            Because of that, we leave state as is
        State: a=[2 1 3 5], n_swaps=1, swap_flag=true EndState
        swap_flag is true, so do another iteration
    Iteration:
        set swap_flag=false. The state is:
        State: a=[2 1 3 5], n_swaps=1, swap_flag=false EndState
        Pair a[1,2] = [2 1] Check if 2<1. Is it true? No.
                            Thus, we set swap_flag=true, increase n_swaps by one,
                            and in the latest a=[2, 1, 3, 5]
                            swap 2 and 1 to get into state:
        State: a=[1 2 3 5], n_swaps=2, swap_flag=true EndState
        Pair a[2,3] = [2 3] Check if 2<3. Is it true? Yes.
                            Because of that, we leave state as is
        State: a=[1 2 3 5], n_swaps=2, swap_flag=true EndState
        Pair a[3,4] = [3 5] Check if 3<5. Is it true? Yes.
                            Because of that, we leave state as is
        State: a=[1 2 3 5], n_swaps=2, swap_flag=true EndState
        swap_flag is true, so do another iteration
    Iteration:
        set swap_flag=false. The state is:
        State: a=[1 2 3 5], n_swaps=2, swap_flag=false EndState
        Pair a[1,2] = [1 2] Check if 1<2. Is it true? Yes.
                            Because of that, we leave state as is
        State: a=[1 2 3 5], n_swaps=2, swap_flag=false EndState
        Pair a[2,3] = [2 3] Check if 2<3. Is it true? Yes.
                            Because of that, we leave state as is
        State: a=[1 2 3 5], n_swaps=2, swap_flag=false EndState
        Pair a[3,4] = [3 5] Check if 3<5. Is it true? Yes.
                            Because of that, we leave state as is
        State: a=[1 2 3 5], n_swaps=2, swap_flag=false EndState
        swap_flag is false, so stop the iteration
Final List: 1, 2, 3, 5
Number of swaps: 2
END OF EXECUTION

Problem: 0, 3, 8, 5, 6
EXECUTION
```

**Prompt 2.** Logical deduction puzzle IRSA prompt. Playground link (use with 0 temperature)

```
PUZZLE: The following objects need to be ordered. obj1 is the biggest. obj2 is smaller than obj3.
obj1 is bigger than obj2. QUESTION: Which object is the biggest?
START
Parsing step:
    Items: obj1, obj2, obj3
    Numbe of items: 3
    Statement 1: obj1 is the biggest.
    Statement 2: obj2 is smaller than obj3.
    Statement 3: obj1 is bigger than obj2.
Scoring identification step:
     Scores will refer to size.
     Since we have 3 items, let's assume that the biggest gets a score  of 3 pounds
     and the smallest gets the score of 1 pound.
Translation step:
    Available variable names: x, y, z, a, b, c
    Map item scores of 'obj1', 'obj2', 'obj3' to variable names x, y, z
    obj1 score is x; obj2 score is y; obj3 is z;
    Statement 1: 'x' is the biggest.
    Statement 2: 'y' is smaller than 'z'.
    Statement 3:  'x' is bigger than 'y'.
Initialization step:
    Words used to qualify the realtionsips: smaller, bigger, biggest
    Orientation step:
        the biggest: refers to the score of 3
        smaller: refers to smaller score
        bigger:  refers to larger score
    Initialize so that all scores are  different numbers between 1 and 3
    Score_assignment_A:
    x=2, y=3, z=1
Iterative reasoning
Iteration 1:
    update_flag=false
    Statement 1:  'x' is the biggest, meaning: x should be 3
    In Score_assignment_A, x is 2
    x is not what it should be, so we need to make a change, so we set update_flag=true and we need to make a swap.
    In the statement there is only one variable and it is  x. We need  to find another. We want x to be 3,
    but we see that in Score_assignment_A that 3 is assigned to y, so we swap  values of x and y to make
    Score_assignment_B:
    x=3, y=2, z=1
    Statement 2: 'y' is smaller than 'z', meaning: y<z
    In Score_assignment_B, y is 2 and z is 1,  so y<z maps to 2<1
    2<1 is false, so we need to make a change, so we set update_flag=true and we  need  ot make a swap.
    In the statement there are two variables and those are y and z so we swap in Score_assignment_B to make
    Score_assignment_C:
    x=3, y=1, z=2
    Statement 3: ' x' is bigger than 'y', meaning x>y
    In Score_assignment_C, x is 3 and y is 1,  so x>y maps to 3>1
    3>1 is true, so we don't need to make a change.
End of iteration. Since update_flag is true, we need more iterations.
Iteration 2:
    update_flag=false
    Statement 1:  'x' is the biggest, meaning: x=3
    In Score_assignment_C, x is 3,  so x=3 maps to 3=3
    3=3  is true, so we don't need to make a change.
    Statement 2: 'y' is smaller than z, meaning: y<z
    In Score_assignment_C, y is 1 and z is 2, so y<z maps to 1<2
    1<2 is true, so we don't need to make a change.
    Statement 3: 'x' is bigger than y, meaning x>y
    In Score_assignment_C, x is 3 and y is 1,  so x>y maps to 3>1
    3>1 is true, so we don't need to make a change.
End of iteration. Since update_flag is false, we have finished all iterations and found the correct order.
The correct score assignment is the last (Score_assignment_C):
x=3, y=1, z=2
Reverse translation step:
Map items 'obj1', 'obj2', 'obj3' to variable names x, y, z
so we replace x by obj1, y by obj2, and z by obj3 to get size scores:
obj1 has the score 3; obj2 has the score 1; obj3 has the score 2

Question: Which object is the biggest?
Answer: obj1
Sorting all by score starting with obj1:
with score 3, obj1
with score 2, obj3
with score 1, obj2
END

PUZZLE: On a shelf, there are five books: a gray book, a red book, a purple book, a blue book, and a black book.
The red book is to the right of the gray book. The black book is to the left of the blue book.
The blue book is to the left of the gray book. The purple book is the second from the right.
QUESTION: Which is leftmost?
START
```

the right of a raven, but a sparrow is the left-most). Despite the low number of objects (e.g., five) in these puzzles, LLMs struggle to solve them in zero- or few-shot settings, much like how human solvers typically cannot just see the correct answer instantly without scratch paper. This task is not solved well by LLMs without external search/reasoning/inference algorithms, such as ThinkSum [22]. However, a variant of BubbleSort algorithm adapted to this problem and shown in Prompt 2 can be used to solve 76% of these puzzles. The prompt has a CoT structure that translates the problem into a canonical form, and then, in IRSA style, describes an iterative swapping procedure that rearranges the objects.

## 2.2 Fragmented prompting.

Another way to trigger iterative behaviour is through fragmented prompting, illustrated in Prompt 3), which relies on:

- **Complete state specification**. In contrast to Prompt 1 where iterative behaviour is induced indirectly through worked-out examples of multiple full loops, Prompt 3 explicitly describes state content in state-to-state transitions, including the iterator $i$.

- **Fragmentation.** Prompt 3 does not fully cover the entire execution path of any single example. Instead, it follows the first three state changes[2] for the sequence $2, 3, 1, 5$, and then stops in the middle of a sentence. Then it shows 6 additional fragments of execution paths for *different* problems.

Interestingly, fragmented prompting can also trigger iterative behaviour, where the language model accurately executes the algorithm on a given input and outputs END OF EXECUTION when the termination condition (no new updates on the sequence) is met. Viewing this prompt as an instance of in-context learning, it is challenging to classify it in usual terms. It goes beyond 0-shot learning as it contains explanations specific to the algorithmic sorting task. Yet, as opposed to what the few-shot CoT prompting might do, it does not work out any single example of array sorting. Instead, it provides fragments of patterns that can be stitched together to execute the algorithm (and GPT-3 CODE-DAVINCI-002 does execute it correctly for new inputs).

The potential advantage of such fragmented prompting is that the prompt can be shorter and include a greater variety of situations that may be encountered in new problems. A potential disadvantage is that the language model may get confused by the fragmentation and start hallucinating new independent fragments. In this case, we managed to avoid that by having the first fragment starting from the start of execution, going through several state transitions, and ending mid-sentence. Because of this, when a new problem is given, the language model starts running the execution path from the beginning, and later refers to various cases in the prompt for guidance on how to proceed.

## 2.3 Skip attention.

Prompt 3 also illustrates the idea of attention skipping. Whether using a single-execution or a fragmented prompt, if the state in the `<state>*</state>` structure is complete, the attention mechanism can generate the next token without attending to all the generated text. **It is sufficient to attend to the prompt and the text generated after and including the last state.**

If the skipping is implemented on the server side, akin to stop word functionality, then skipping unnecessary attention saves computation: The state of the model at the end of the prompt is cached and used to continue processing from the latest generated `<state>` marker, ignoring the text generated in-between. Skip-to-state can also be implemented on the client side, iteratively updating the original prompt by concatenating the latest `<state>*</state>` structure to the original prompt and calling the generative model with `</state>` as a stop sequence (We did the latter in our experiments). In both cases, the skip-to-state strategy should increase the number of tokens that can be generated, as self-attention, which grows linearly with the generated text, is the primary cause for the token limitations. Skip-to-state strategy keeps the self-attention cost constant. As IRSA requires the unrolling of potentially long iterative algorithms, these savings are important. For example, running a dynamic program that keeps track of 2D matrices is only practical in this manner. (See also [29] on an external memory approach to dealing with limited attention length. Here we deal with it by skipping parts of generated text, instead). Another advantage of skip-to-state attention is that by only attending to the necessary information, the generative model is less likely to get confused by accidental patterns created in its own generated text. (See more in Section A.3 and Figure A.1.)

---

[2]The full execution path in this style is shown in Prompt A.4.

**Prompt 3. Fragments**: An incomplete path for the first few Buble Sort state transitions for one sequence is followed by state transitions involving *different* sequences at *different* execution points. Initial part of the response is marked green. **Skip attention**: The part of the response up to the last state is not needed to continue the generation. Only the prompt, the last `<state>*</state>`, and the text after it are necessary to generate the next token. Playground link (use with 0 temperature)

```
Problem: 2, 3, 1, 5
EXECUTION
    Length of the list: L=4
    Number of pairs: P=3
    a=[2 3 1 5]
    set n_swaps=0. set i=P=3. set swap_flag=true.
        <state> a=[2 3 1 5] i=3 P=3 n_swaps=0 swap_flag=true </state>
    Since i=3 and P=3, i and P are equal, so this iteration is done, but swap_flag is true,
    so we need another iteration
    Iteration:
        set swap_flag=false.  set i=0. The state is:
        <state> a=[2 3 1 5] i=0 P=3 n_swaps=0 swap_flag=false </state>
        Since i=0 and P=3, these two are different, so we continue
        a[i]=a[0]=2 a[i+1]=a[1]=3
        Because 2<3 is true we keep state as is and move on by increasing i
        <state> a=[2 3 1 5] i=1 P=3 n_swaps=0 swap_flag=false </state>
        Since i=1 and P=3, these two are different, so we continue
        a[i]=a[1]=3 a[i+1]=a[2]=1
        Because 3<1 is false we set swap_flag=true,increase n_swaps by one, and in a=[2 3 1 5]
        swap 3 and 1, and increase i, and keep P as is to get
        <state> a=[2 1 3 5] i=2 P=3 n_swaps=1 swap_flag=true </state>
        Since i=2 and

        <state> a=[6 5 8 9 1 2] i=2 P=5 n_swaps=5 swap_flag=false </state>
        Since i=2 and P=5 i and P are different, so we continue
        a[i]=a[2]=8 a[i+1]=a[3]=9
        Because 8<9 is true we we keep state as is and move on by increasing i
        <state> a=[6 5 8 9 1 2] i=3 P=5 n_swaps=5 swap_flag=false </state>

        <state> a=[9 1] i=0 P=1 n_swaps=2 swap_flag=true  </state>
        Since i=0 and P=1 i and P are different, so we continue
        a[i]=a[0]=9 a[i+1]=a[1]=1
        Because 9<1 is false we set swap_flag=true,increase n_swaps by one, and in a=[9 1] swap 9 and 1
        and increase i, and keep P as is to get
        <state> a=[1 9] i=1 P=1 n_swaps=3 swap_flag=true  </state>

        <state> a=[6 7 3 5] i=3 P=3 n_swaps=7 swap_flag=false </state>
        Since i=3 and P=3 i and P are equal, so this iteration is done, swap_flag is false, so stop
Final List: 6, 7, 3, 5
Number of swaps: 7
END OF EXECUTION
        <state> a=[3 5 6 8] i=3 P=3 n_swaps=1 swap_flag=true  </state>
        Since i=3 and P=3 i and P are equal, so this iteration is done, but swap_flag is true,
        so we need another iteration
    Iteration:
        sset swap_flag=false.  set i=0. The state is:
        <state> a=[3 5 6 8] i=0 P=3 n_swaps=1 swap_flag=false </state>

        <state> a=[2 8 1 3 5 7 4] i=1 P=6 n_swaps=5 swap_flag=false </state>
        Since i=1 and P=6 i and P are different, so we continue
        a[i]=a[1]=8 a[i+1]=a[2]=1
        Because 8<1 is false we set swap_flag=true,increase n_swaps by one, and in a=[2 8 1 3 5 7 4]
        swap 8 and 1 and increase i, and keep P as is to get
        <state> a=[2 1 8 3 5 7 4] i=2 P=6 n_swaps=6 swap_flag=true  </state>

        <state> a=[4 8] i=0 P=1 n_swaps=7 swap_flag=true  </state>
        Since i=0 and P=1 i and P are different, so we continue
        a[i]=a[0]=4 a[i+1]=a[1]=8
        Because 4<8 is true we we keep state as is and move on by increasing i
        <state> a=[4 8] i=1 P=1 n_swaps=7 swap_flag=true  </state>
Problem: 3, 1, 8, 9, 6
EXECUTION
```

```
    Length of the list: L=5
    Number of pairs: P=4
    a=[3 1 8 9 6]
    set n_swaps=0. set i=P=4. set swap_flag=true.
        <state> a=[3 1 8 9 6] i=4 P=4 n_swaps=0 swap_flag=true </state>
        Since i=4 and P=4 i and P are equal, so this iteration is done, but swap_flag is true,
        so we need another iteration
    Iteration:
        set swap_flag=false.  set i=0. The state is:
```

```
        <state> a=[3 1 8 9 6] i=0 P=4 n_swaps=0 swap_flag=false </state>
        Since i=
```

### 2.4 GPT as a machine language: Prompting to interpret/compile a program.

A general-purpose computer can execute algorithms that convert the text of a program into its machine code. Analogously, we can design prompts with instructions on how to turn code in some language into execution paths that can then be used in prompting. The details are given in Section A.1 in the Appendix. In fact, we used a "GPT compiler" in Prompt A.2 to create an execution path for the double loop DP algorithm for finding the longest common subsequence (LCS) which we used in our experiments.

## 3 Experiments

We evaluated two versions of iteration by regimenting self-attention (IRSA):

- **Basic IRSA**: Prompting with highly structured single execution path examples (Table 1). Although similar to CoT prompting, there are notable differences. CoT prompts typically provide multiple steps of reasoning shown for a few examples and have the LLM perform the same steps on a new example. Conversely, IRSA prompts are designed to trigger iterative reasoning that is repeated until the stop condition is reached and the solution is found. Furthermore, the execution path example for each task is deliberately chosen to be out-of-distribution (e.g., the Bubble Sort prompt features a worked-out example of sorting a four-number sequence in just three passes, while the dataset consists of five-number sequences requiring 2 to 5 iterations and up to 20 state transitions, with varying complexity across problems). Thus in terms of information they provide, these prompts can be seen as somewhere between single-shot and zero-shot prompts.
- **Skip-to-state IRSA**: Prompting as above, but with additional forced attention skipping. In this approach, the LLM is forced to attend only to the prompt and the last generated state as it iterates through the input to find the solution (illustrated at the end of Prompt 3). We also evaluate fragmented prompts (Table 2), where the prompt does not consist of a single complete execution path for an example, but instead shows several state-to-state transitions for different inputs.

**Baselines.** To make fair comparisons and avoid unnecessary recomputation, we reused existing baselines from [32] wherever possible, denoted by an asterisk (*) (especially considering that these baselines typically perform close to random guessing on certain tasks). We reused these datasets and baselines for the following tasks: Logical deduction, Balanced parenthesis, and Longest common subsequences for long sequences. We created our own datasets and ran baselines for the following tasks: Bubble sort, Longest substring without repeating characters, and Longest common subsequence for short sequences. We include the best result from [32] for the GPT family, as our experiments were mainly conducted using GPT-3. Our baselines included zero or few (5) shot prompting with or without relevant code added to the description of the task in the prompt (e.g. Prompt A.9). Few shot baselines were made with 5 different random choices of examples to be included in the prompt. The 'Guessing' strategy refers to picking the most frequently correct answer for a given task as a guess for each problem in the task, which is different from truly random guessing. Few-shot prompting could prime the answers to pick the most frequently seen answer, even when no understanding of the problem occurs, which makes our 'Guessing' strategy more reflective of the task difficulty.

**Models.** We have briefly experimented with different members of the GPT-3 family, but ran complete experiments with CODE-DAVINCI-002 for two reasons: TEXT-DAVINICI-002 and 003 often produced qualitatively similar results, and experimentation with the CODE-DAVINCI-002 was easier due to better combination of token quota and availability. Having been tuned on code, this model may have slight advantages over models tuned for more natural language tasks. Nevertheless, as we show in the experiments and discuss in Section A.3, without IRSA, CODE-DAVINCI-002 cannot solve the problems discussed here, even when it can generate the code that could. To induce iterative reasoning in LLMs, it appears that attention needs to be highly regimented through strong structure, and possibly additional attention control, such as the skip-to-state strategy we described in Section 2.3. This also applies to GPT-4 [21]: In Section A.3.3 in Appendix, we show that prompting GPT-4 with straight-forward Prompts A.10, A.11, A.12 does not match the performance of IRSA in GPT-3.

**Datasets.** To test our proposed methods with various prompting baselines, we focus on challenging programming tasks including computer science algorithms from the school curricula and coding interviews for software engineers as follows.

**Bubble sort.** We created a dataset of 100 random non-repeating digit sequences of length 5. For each sequence, we ran the bubble sort algorithm to establish the total number of element swaps it requires. The task is to predict the number of swaps for a given sequence.

Table 1: Iteration through Regimented Self-Attention (IRSA) compared with standard in-context learning baselines, and with the strategy of always guessing the most frequent answer. (*) denotes the best result for GPT-3 from the BIG-bench [32].

| Task | IRSA | Baseline | Guessing |
|------|------|----------|----------|
| Bubble sort | | | |
|   - Prompt 1 | 0.74 | 0.27 | 0.23 |
|   - Prompt A.4 | 1.00 | 0.27 | 0.23 |
| Longest substring | 1.00 | 0.60 | 0.59 |
| Logical deduction | 0.76 | $0.32^*$ | 0.2 |
| Parentheses | 0.96 | $0.56^*$ | 0.5 |

**Longest substring without repeating characters.** A classical coding interview question: Given a string of letters, find the longest contiguous substring such that no letter appears more than once. We created a dataset of 100 random strings of length 7, and for each found the length of the longest subsequence without repeating characters. The task is to predict that length for a given sequence.

**Logical deduction [32].** We include this task (Section 2.1) in experiments to emphasize the broad importance of triggering iteration in LLMs responses. Enabling LLMs to execute iterative algorithms through effective prompting could help solve numerous reasoning problems. In particualr, this task that involves solving a puzzle about an order of items/objects/persons, such as books on the shelf, birds on a branch, cars, golfers, etc., given several clues, such as "minivan is more expensive than the car", or "the robin is to the left of the finch." We focus on a subtask involving 5 items, with varying sets of items and the types of ordering across the puzzles. While in-context learning with LLMs consistently solves less than $35\%$ of puzzles, a recent combination of GPT-3 and probabilistic reasoning [22] was able to solve $77\%$ of the puzzles. We reach a similar performance through IRSA, *without* an additional external reasoning mechanism.

**Valid parentheses [32].** The task is the first of the two in the cs-algorithms challenge in BIG-bench. The goal is to evaluate LLMs ability to perform reasoning equivalent to the classical stack manipulations needed to check if a sequence of parentheses of different types is balanced. LLMs (including GPT) tend to do about the same as chance ($50\%$), except for PaLM with 3 shots, which gets around $75\%$ accuracy.

**Longest common subsequence (long) [32].** The second task in BIG-bench cs-algorithms involves solving the classical dynamic programming problem. Defining a subsequence of a sequence to be a sequence of symbols one could get by skipping arbitrary stretches in the original sequence, the task is to find the length of the longest subsequence common to two given sequences. LLMs do not do much better than chance on this task ($\sim 10\%$).

**Longest common subsequence (short).** We created this dataset in the same manner as the above one from the BIG-bench, but with the constraint on the sequence lengths, limiting them to a maximum of 6 characters. This allows us to evaluate IRSA on more cases, ensuring it does not run out-of-memory (tokens) in generation[3].

**Basic IRSA results.** The basic IRSA results are summarized in Table 1. For Bubble Sort evaluations, we show the results using both Prompt 1, and Prompt A.4. The latter is a single execution path for the same problem $(2, 3, 1, 5)$, but in the style of Fragmented Prompt 3 by continuing the execution path initiated by Prompt 3, without incorporating fragments from other paths. The former had an accuracy of $74\%$ for inferring the numbers of swaps necessary to sort different sequences, while the latter achieved $100\%$. Note that while the execution path for the example $2, 3, 1, 5$ requires three iterations of the outer loop and three iterations in each inner loop, the dataset contains sequences of length 5 and thus requires four iterations in the inner loop and a variable number of iterations of the outside loop – anywhere from 2 to 5 – and yet the model can execute the correct number of iterations based on the stoppage criterion (that in the inner loop, no changes were made to the sequence).

For the logical deduction puzzles, we used the Prompt 2 Appendix. Note that the logic of the iterative reasoning there is faulty as it may enter an infinite loop. When that happens, the generation runs out of tokens and we simply used the answer after the 4th iteration in evaluation. Further

---

[3]Buble sort, Longest substring, and LCS (short) datasets: https://github.com/anajojic/gpt-coding

Table 2: IRSA with skip-attention, Bubble Sort and Longest Common Subsequence problems. Fragmented prompting, Bubble Sort problems. (*) denotes the best GPT result in [32]

| Baselines | Bubble Sort | LCS-S | LCS-L |
|---|---|---|---|
| 0-shot | 0.20 | 0.09 | $0.14^*$ |
| 0-shot + code | 0.20 | 0.11 | - |
| few shot | $0.25_{\pm 0.05}$ | $0.07_{\pm 0.01}$ | $0.16^*$ |
| few shot + code | $0.23_{\pm 0.03}$ | $0.06_{\pm 0.02}$ | - |
| Guessing | 0.23 | 0.44 | 0.10 |
| **IRSA skip-to-state** | | | |
| single path | 0.95 | 0.93 | 0.28 |
| 7 fragments | $0.99_{\pm 0.02}$ | - | - |
| 13 fragments | $0.97_{\pm 0.03}$ | - | - |
| 19 fragments | $0.99_{\pm 0.02}$ | - | - |
| 25 fragments | $0.97_{\pm 0.03}$ | - | - |

discussion in Section A.3 suggests the potential for creating more effective prompts. Nevertheless, with this prompt to induce iterative reasoning, we still reach the state-of-the-art results, comparable only with [22], which uses an external reasoning mechanism in conjunction with prompting. To solve the longest substring without repeating characters problems, we developed Prompt A.5 based on asingle-pass algorithm (Section A.2), which, interestingly, trades computation for memory. To address the parentheses problem, we used the single execution path that demonstrates stack operations for determining whether the sequence is balanced or not. The beginning and the end are shown in Prompt A.6 and discussed in Section A.2.1 in the Appendix.

**Skip-to-state attention results.** The dynamic programming solution to the longest common subsequence (LCS) problem requires its state including a $(M+1) \times (N+1)$ matrix storing the solution for all prefixes of the two sequences of lengths $M$ and $N$. Without skip-to-state attention (Section 2.3), the API calls run out of tokens before reaching the end for all but the shortest problems. Using the approach described in Section 2.4, A.1, we compiled an execution path in Prompt A.3, and then used it to induce IRSA on LCS short (LCS-S) and LCS long (LCS-L) problems. Even with skip attention, the state was too large to fit the token limit for most of the problems in LCS-L from BIG-bench. Yet, IRSA with skip attention still beats the state-of-the-art significantly (Table 2). On shorter problems in LCS-S, where IRSA with skip-attention does not run out of tokens, the performance was a respectable 93%. Note that even GPT-4, without IRSA, can only reach 69% accuracy on LCS-S (Section A.3.3).

We tested **fragmented prompting** of Bubble Sort execution (Table 2). For each selected number of fragments – 7, 13, 19, 25 – at least one of five randomly generated prompts achieved 100% accuracy. These prompts followed the format in Prompt 3, starting with the few state transitions from the beginning for the sequence $[2, 3, 1, 5]$ and then listing additional 6, 12, 18, or 24 fragments. Bubble Sort has 6 different transitions, and fully balanced instruction listing one, two, three, or four of each type, with a random sequence in the state, leads to a slightly better performance than having a completely randomly chosen execution path fragments. These six basic transitions, illustrated in Prompt 3, involve two ways of ending an iteration depending on the swap flag and four ways of changing the state: two possibilities for inequality being true or not, combined with two possible previous values of the swap flag. We found that the prompt sensitivity causes different prompts to fail for different test cases: Each of the fragmented prompt collections yields 100% as an ensemble.

## 4 Conclusion

We demonstrated that GPT-3 can be triggered to execute iterative algorithms, including double loops, with variable termination conditions. This has several consequences discussed in Appendix (Section A.3). For example, IRSA may find applications in sofware engineering and education. If LLMs are Turing Machines (in addition to being natural language translators and analyzers), their evaluation probably needs to be rethought, esp. in cases where models are expected to make inferences for which we have algorithms, because in-context learning would cover prompts designed to execute them (Section A.3). Regimenting self-attention for a given task may require a different level of effort (Section A.3.2, but even GPT-4 cannot execute programs consistently without IRSA (Section A.3.3).

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
