# A Appendix

## A.1 GPT as a machine language: Prompting to interpret/compile a program.

A general-purpose computer can execute algorithms that convert the text of a program into its machine code. Analogously, we can design prompts with instructions on how to turn code in some language into execution paths that can then be used in prompting.

An example is shown in Prompt A.2 (Appendix), where several examples of hypothetical syntax for transforming states are given, including setting values of variables and matrices, printing them, a single loop program execution, and the `detailed_max` function that breaks down steps and explains them. Then, the double loop dynamic programming algorithm for finding the longest common subsequence (LCS) is also presented in this new language. This prompt successfully triggers the correct execution of the algorithm, complete with detailed explanations and state transitions (green shaded in Prompt A.3). This can then be used as a prompt to execute the LCS algorithm on arbitrary inputs (Section 3). We should note that GPT-3 is still sensitive to small alterations in text, and Prompt A.2 does not always lead to good interpretations of the algorithm. The performance may depend on accidental deceptive patterns and inconsistencies in the prompt, as well as the input. Nevertheless, once the output has been verified as correct, the Prompt A.2 together with the response in Prompt A.3 became the prompt – IRSA 'machine code' for GPT — to execute (mostly correctly) the LCS algorithm for new inputs, as long as they are appended in the same format:

```
LCS:
Input: <seq1> <seq2> End of input
LCS Prep:
```

## A.2 The longest substring without repeating characters

To solve the longest substring without repeating characters problems with basic IRSA, we developed Prompt A.5 based on the 1-index version of the following single-pass algorithm. Interestingly, this algorithm trades computation for memory by creating one variable per unique letter in the sequence for storing the location where the letter was last seen in the sequence during the pass (`last_ind`):

```
# s contains the given string
last_ind = {}
m_len = 0

# window start
st_ind = 0

for i in range(0, len(s)):
  if s[i] in last_ind:
    st_ind=max(st_ind,last_ind[s[i]]+1)

  # Update result if window is longer
  m_len = max(m_len, i-st_ind + 1)

  # Update last index of the character
  last_ind[s[i]] = i
return m_len
```

### A.2.1 Balanced parentheses

To address the parentheses problem, we used the single execution path that demonstrates stack operations for determining whether the sequence is balanced or not. The beginning and the end are shown in Prompt A.6. For brevity, we have omitted certain portions (represented by ellipses). Note that creating long prompts is made easier by GPT's completion capabilities, i.e., by starting with a description of a few steps and asking the model to finish it. Wherever we want the prompt to differ from the model's guess, we erase the generated text from that point and continue typing our correction/instruction and try to autocomplete again. (See also Sections A.3, A.1 in the Appendix). But interestingly, as discussed in Section 2.2 on fragmented prompting, parts of the execution paths

can be omitted: Prompt A.6 as is, with the ellipsis instead of 10 steps in the algorithm, still achieves 91% accuracy!

## A.3 Full discussion section

Iteration by Regimenting Self-Attention (IRSA) is a technique for *triggering code execution in GPT-3 models*. Note that the goal is different from the goal of Alphacode [13] and Copilot [4, 25], which are meant to *write* the code, without necessarily understanding what it outputs. While there are indeed examples of rather impressive code generation and even, anecdotally, execution path generation using minimal prompting in the latest Codex and GPT-3 models, the lack of control in current LLMs prevents the consistent achievement of these feats with precision, which is why the code generation applications involve humans in the loop. For instance, as illustrated in zero-shot bubble sort code Prompt A.8, when relying on Codex alone to attempt code execution, the generated samples are intuitively close to the correct solution, but a bit off, preventing correct execution. IRSA, on the other hand, can produce consistently accurate outputs.

In algorithm design, trading computation for memory use is a recurrent idea. IRSA as a technique for LLM inference can be seen in a similar light: We could train a bigger model on more data, with attention spanning deeper into the past tokens, hoping that it could answer a simple yet computationally complex query in just a couple of tokens directly; or we can devise a prompting strategy instructing a smaller LLM to use its token stream as a memory tape, allowing it to reach similar functionality with increased token usage. By triggering and controlling iterative behaviour, we can, in principle, execute arbitrary algorithms, which further raises interesting questions: What are the consequences of LLMs becoming Turing-complete? And how difficult is it to program via IRSA? Will larger GPT models become capable of executing programs correctly without IRSA? Based on our experience in designing the prompts we showed here, we speculate on these three questions in this section.

### A.3.1 Possible consequences

**(Teaching) Coding.** The integration of LLMs' code generation capabilities with IRSA leads to innovative applications in code generation. Some of it is implied in the interpreter/compiler Prompt A.2, which instructs GPT how to interpret and execute code. Following these ideas, exploring program verification and automatic debugging could be a promising direction. Another obvious application of IRSA is in computer science education, where we often expect students to execute programs on paper to determine what the state will be at some point during the execution. Furthermore, IRSA may also point to new ways of programming by example.

**Adversarial applications.** Any time a computational medium is Turing-complete, a variety of malicious uses may become possible, such as creating and executing malware, exploiting system vulnerabilities, conducting cryptographic attacks, causing resource exhaustion, etc. Thus we should be aware of the double-edged sword with the increased versatility and computational power of GPT models.

**In-context learning and LLM evaluation.** Prompting with IRSA must be considered a zero- or one-shot learning technique, analogous to chain-of-thought prompting. If, via IRSA, LLMs can be disciplined with a regimented prompt to execute arbitrary algorithms involving (double) loops, they may be able to solve arbitrary problems NLP researchers can compose, incorporating natural language understanding and iterative reasoning like belief propagation, constraint satisfaction, search, etc. This renders many of the hard BIG-bench tasks easier than they initially appear, as already suggested by [33] using classical CoT prompting. Many CoT results can be further improved with IRSA (as logical deductions with Prompt 2).

However, triggering such iterative behaviour may still be hampered by the same sensitivity of in-context learning to accidental misleading patterns, already observed in classical prompting [16, 41], where there may exist a "fantastical" crafting of the prompt that significantly improves the accuracy of the task. In fact, iterative reasoning may further amplify the fantastical choices. Thus, if one LLM successfully solves a hard logical reasoning task using a suitable prompt while another does not, this might imply that the optimal prompt has not yet been found. In fact, it would not be surprising if better prompts are eventually found that enable the LLM we used here (GPT-3, CODE-DAVINCI-002) to solve all tasks with 100% accuracy. Thus, evaluating LLMs on their in-context learning abilities is

of questionable value: Some of the hard tasks in BIG-bench may be better suited to evaluating the skills of prompt engineers rather than the LLMs themselves.

**Hybrid models – LLMs as translators.** If LLMs are Turing-complete and can transform problems described in natural language into algorithmically solvable programs, the decision to let them execute the program or not becomes a practical matter of computational cost. With the apparent magic of savant-like guessing gone, it is much more practical to run the algorithms on a classical computer, an approach taken by, for example, [22] where the external computational mechanism performs probabilistic inference, or [10] that involves external control flows, and many other recent published and unpublished experiments combining LLMs with external calls and tools [24, 7, 39, 26, 28, 23]. Such hybrid models could separate the higher level reasoning "System 2" – to use an analogy with models of human cognitive processes [35, 9] – from the lower-level "knee-jerk reaction" reasoning "System 1", however savant-like it might be. In such systems, LLMs can dramatically improve traditional artificial intelligence algorithms simply by translating the problems into an appropriate form: see Prompt A.7 where the logical deduction task is solved by creating a call to the `Solve` command in Wolfram language (Mathematica) for an example. The artificial intelligence community is increasingly interested in researching such systems, e.g., [1, 8], and the developer community is already developing and deploying hybrid language models (Bing-ChatGPT integration, for instance).

**Self-attention control in training and inference.** To paraphrase an old adage on parenting, researchers have spent a lot of effort teaching GPTs to pay attention to everything in the text, and now IRSA is an attempt to stop it from attending to everything. We accomplish it both by drawing attention with a strong repetitive structure and by brute force through skip attention (Section 2.3). More flexible ways of determining what the model should attend to may be needed both in model building and inference.

### A.3.2 Pitfalls of programming in GPT-3

Prompts we experimented with induce single loop or double loop program execution. Generally, controlling double loop algorithms, such as Bubble Sort and Longest Common Subsequence, is more challenging. The difficulty lies not in understanding the double loop logic, but rather in the increased probability of running into some of the problems described below. These problems are not always obvious, but can result in a wide range of accuracies achieved by seemingly similar prompts. For example, the two prompt designs for Bubble Sort both worked surprisingly well, but showed a big gap in performance between them ($74\%$ and $100\%$). Here are some tips for attempting IRSA.

**Keep a complete state.** While it is often possible to instruct by analogy without fully accounting for all decisions, keeping the full state (i.e., showing it repeatedly after each transition) is usually preferable. For example, Prompt 3 contains the iterator variable in the state, while Prompt 1 does not. Not only does keeping full state help regiment the attention, but it makes fragmented prompting and skip-to-state attention possible.

**Explain why before the instruction, not after.** LLMs are autoregressive, which makes them easier to prompt in order: from left to right. Thus, instead of instructing with 'We now swap 4 and 2 because 2<4', we instruct with:

`Because 4<2 is false we swap 4 and 2`

Then later in generation, e.g., 'Becasue 5<3 is' will trigger generation of token `false` and it, in turn, will trigger generation of 'we swap', and so on.

**Avoid unnecessary variation, follow strong structure.** We used the term *regimenting* attention in the naming of the technique to emphasize that strong structure is even more important in IRSA than in other prompting applications. It is usually crucial to order the variables in the state always in the same order, utilize the same keywords to designate the state, use the same language to explain the transitions, and ensure consistent capitalization, punctuation, and even spacing/tabulation. We experimented with several variants of the Bubble Sort prompt, and even when using the same worked-out example, the accuracy can vary dramatically.

**Generate as much of the prompt with LLM itself.** One way to create such a strong structure is to let the model continue the prompt we are designing after every few lines (going back to correct the incorrectly generated continuation). The model is more likely to stay faithful to the pattern human started than the human is (with spacing, typos, and so on). Because of this, using the

interpreter/compiler Prompt A.2 to create an LCS execution path to serve as a prompt is a safer way of generating an IRSA-inducing prompt (as long as we verify that the exemplary execution path is correct).

**Overlapping patterns can be problematic.** When generating the next token, an LLM has to balance many influences of patterns both in the prompt and the so-far generated text. For example, in the LCS algorithm execution Prompt A.3, the model has to balance the long-range self-attention when deciding the next token after `C[1,1]=` with the short-range influences, which make the token 1 most likely after two 1s in a row regardless of the longer context. At times, short-range influences prevail and cause an incorrect execution. But, long-range self-attention can also inappropriately overrule correct short-range reasoning. For instance, when generating based on the Bubble Sort Prompt 3, the model generates repetitive text that includes many statements of the form 'Because $n<m$ is true/false ...,' which can create strong pattern overruling local evaluation of the next inequality. To demonstrate that, we evaluated the likelihood of the next token after 'Because 2<1 is' for different lengths of context preceding this text. The context had between 1 and 15 lines of text in the form 'Because 2<$m$ is true we ...' with $m \in [3..9]$ randomly chosen, e.g.

```
Because 2<3 is true we ...
Because 2<7 is true we ...
Because 2<5 is true we ...
Because 2<1 is
```

As we show in Fig A.1, although the preceding context is correct when evaluating the inequalities, the log odds of an incorrect evaluation of 2<1 increase by over six orders of magnitude with the length of this context. The longer this context is, the more it reinforces the pattern 'Because 2< ... true': If 2 was smaller than a variety of numbers, then it is smaller than 1, too! Furthermore, there is a large variation due to the random selection of $m$ in the examples in the context, indicating a variety of other patterns that drive the generation (The figure shows the band between the maximum and minimum log odds over 20 runs). For the contexts of length 7 the odds of picking `true` over `false` become roughly even. IRSA can drive probabilities to be so taut that rerunning the same API call with zero temperature can sometimes return a different result (The code behind the API presumably always adds a very small constant to log probabilities before sampling). Skip-to-state strategy in Section 2.3 is thus less sensitive to patterns that result from program execution.

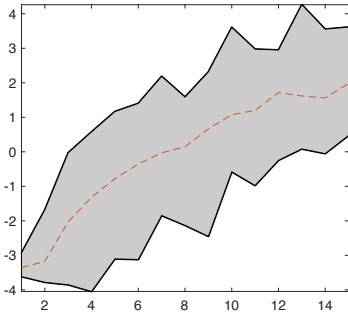

Figure A.1: The difference between GPT Codex log probabilities of tokens `true` and `false` after 'Because 2<1 is', which was preceded by a long context of variable length (x-axis). The context contains between 1 and 15 lines of text comparing number 2 with randomly chosen *larger* numbers and declaring, e.g., Because 2<6 is true ... We show the band between the maximum and minimum log odds over 20 trials, as well as the mean of the difference. When the preceding context does not have too many comparisons of 2 with larger numbers, the model overwhelmingly prefers the correct evaluation `false`, but when the context is longer than 7 statements, the model usually prefers `true`.

This fragility further emphasizes the difficulty in evaluating LLMs on in-context learning tasks: Improving accuracy may simply be a matter of spending more time designing a prompt (becoming a GPT whisperer). Still, getting GPT to execute the algorithms studied here was not excessively hard, and it may even become easier on newer models.

### A.3.3 And what about GPT-4?

A recent qualitative analysis of GPT-4 abilities [3] includes one example of detailed execution of a Python program for one input (in their Fig. 3.7). The LCS algorithm is well-known, so would the newer and better GPT-4 model execute it correctly and consistently across different inputs? In Prompt A.10, we show a prompt that simply asks GPT-4 to show the LCS algorithm, execute it, and report the result. On our LCS-S dataset, using this prompt design and sampling with zero temperature, GPT-4 gets the correct answer 49% of the times, just slightly better than the 'Guessing' baseline (Table 1). An alternative prompt shown in Prompt A.11, asks for intermediate steps of execution to be shown before the answer is generated, moving the prompting strategy closer to IRSA. This prompt can be thought of as a version of Prompt A.2, but lighter and more straightforward, expecting GPT-4 to be able to show program execution without strict specifications. This prompt leads to the accuracy of 69% on LCS-S, still behind IRSA result with codex (93%, Table 2). To illustrate why this may be, in Prompt A.12 we show the same prompt asking for intermediate steps, but for a different input. The inputs in Prompts A.11 and A.12 were processed differently, even though everything else in the prompts was the same, and API calls were made with zero temperature. In one case, only the initial and end states of the "dp" matrix are shown, while in the other, several steps (but not all!) are shown. Therefore, it seems that GPT-4 is still hard to control without regimenting self-attention more strictly.

### A.4 Full set of prompts

Here we list the prompts used in all tasks and discussed above. The caption of each prompt contains a saved link to the OpenAI playground.

**Prompt A.1.** Response (shaded green) to Prompt 1, whose ending part is shown.

```
...
        State: a=[1 2 3 5], n_swaps=2, swap_flag=false EndState
        swap_flag is false, so stop the iteration
Final List: 1, 2, 3, 5
Number of swaps: 2
END OF EXECUTION

Problem: 0, 3, 8, 5, 6
EXECUTION
```

```
    Prep
    Length of the list: 5
    Number of consecutive pairs: 4
    a=[0 3 8 5 6]
    set n_swaps=0
    EndPrep
    Iteration:
        set swap_flag=false. The state is:
        State: a=[0 3 8 5 6], n_swaps=0, swap_flag=false EndState
        Pair a[1,2] = [0 3] Check if 0<3. Is it true? Yes.
                        Because of that, we leave state as is
        State: a=[0 3 8 5 6], n_swaps=0, swap_flag=false EndState
        Pair a[2,3] = [3 8] Check if 3<8. Is it true? Yes.
                        Because of that, we leave state as is
        State: a=[0 3 8 5 6], n_swaps=0, swap_flag=false EndState
        Pair a[3,4] = [8 5] Check if 8<5. Is it true? No.
                        Thus, we set swap_flag=true, increase n_swaps by one,
                        and in the latest a=[0 3 8 5 6]
                        swap 8 and 5 to get into state:
        State: a=[0 3 5 8 6], n_swaps=1, swap_flag=true EndState
        Pair a[4,5] = [8 6] Check if 8<6. Is it true? No.
                        Thus, we set swap_flag=true, increase n_swaps by one,
                        and in the latest a=[0 3 5 8 6]
                        swap 8 and 6 to get into state:
        State: a=[0 3 5 6 8], n_swaps=2, swap_flag=true EndState
        swap_flag is true, so do another iteration
    Iteration:
        set swap_flag=false. The state is:
        State: a=[0 3 5 6 8], n_swaps=2, swap_flag=false EndState
        Pair a[1,2] = [0 3] Check if 0<3. Is it true? Yes.
                        Because of that, we leave state as is
        State: a=[0 3 5 6 8], n_swaps=2, swap_flag=false EndState
        Pair a[2,3] = [3 5] Check if 3<5. Is it true? Yes.
                        Because of that, we leave state as is
        State: a=[0 3 5 6 8], n_swaps=2, swap_flag=false EndState
        Pair a[3,4] = [5 6] Check if 5<6. Is it true? Yes.
                        Because of that, we leave state as is
        State: a=[0 3 5 6 8], n_swaps=2, swap_flag=false EndState
        Pair a[4,5] = [6 8] Check if 6<8. Is it true? Yes.
                        Because of that, we leave state as is
        State: a=[0 3 5 6 8], n_swaps=2, swap_flag=false EndState
        swap_flag is false, so stop the iteration
Final List: 0, 3, 5, 6, 8
Number of swaps: 2
END OF EXECUTION
```

**Prompt A.2.** Interpreter/Compiler prompt that can execute LCS algorithm. Playground link (use 0 temperature)

```
Matrix C contains values, e.g:
C[0,0]=1 C[0,1]=6 C[0,2]=11 C[0,3]=16 C[0,4]=21
C[1,0]=2 C[1,1]=7 C[1,2]=12 C[1,3]=17 C[1,4]=22

To query a value:
Show(C[1,2])
<state> C[1,2]=12 </state>

Query an undefined variable
Show(a)
<state> a=0 </state>

To set a value:
C[0,1]:=8
C[0,1] was 6. Now C[0,1]=8.
a:=5
a was 0. Now a=5.

To query multiple variables:
Show(a, C[0:1,0:4])
<state>
a=5
C[0,0]=1 C[0,1]=8 C[0,2]=11 C[0,3]=16 C[0,4]=21
C[1,0]=2 C[1,1]=7 C[1,2]=12 C[1,3]=17 C[1,4]=22
</state>

Program:
N:=1
for i from 0 to N
    C[i,i]=-3
Execute:
i:=0
C[0,0]:=-3
i:=1
C[1,1]:=-3
Finished with i=N, so done with i loop

Show(C)
<state>
C[0,0]=-3 C[0,1]=8 C[0,2]=11 C[0,3]=16 C[0,4]=21
C[1,0]=2 C[1,1]=-3 C[1,2]=12 C[1,3]=17 C[1,4]=22
</state>

Program:
if a<C[0,4]
    C[0,0]:=5
else
    C[0,1]:=2
Execute:
Check if a<C[0,1]?  a is 5 C[0,1] is 8 Is 5<8?...
  ... Yes. C[0,0]:=5
Done

Show(C[0,0])
<state> C[0,0]=5 </state>

Program:
C[0,2]:=detailed_max(C[0,3],C[0,4])
Execute:
C[0,3] is 16, C[0,4] is 21. C[0,2] is the greater of...
  ...them. C[0,2]:=21
Done

LCS:
Input: TA ATA End of input
LCS Prep:
a[1]=T a[2]=A
b[1]=A b[2]=T b[3]=A
M=2 N=3

LCS program:
Show(a,b,M,N)
for i from 1 to M
    for j from 1 to N
        if a[i]==b[j]
            C[i,j]:=C[i-1,j-1]+1
        else
            C[i,j]:=detailed_max(C[i,j-1],C[i-1,j])
        Show(i, j, M, N, C[0:i,0:N])
Show('END')
Execute:
```

**Prompt A.3.** In response to Prompt A.2, the model returns the correct LCS execution path shown here
...

```
Execute:
```

```
<state>
a[1]=T a[2]=A
b[1]=A b[2]=T b[3]=A
M=2 N=3
</state>
i:=1
j:=1
Check if a[1]==b[1]?  a[1] is T b[1] is A Is T==A?...
   ... No. C[1,1]:=detailed_max(C[1,0],C[0,1])
   ... C[1,0] is 0, C[0,1] is 0. C[1,1] is the greater of
   ...them. C[1,1]:=0
<state>
i=1 j=1 M=2 N=3
C[0,0]=0 C[0,1]=0 C[0,2]=0 C[0,3]=0
C[1,0]=0 C[1,1]=0 C[1,2]=0 C[1,3]=0
</state>
j:=2
Check if a[1]==b[2]?  a[1] is T b[2] is T Is T==T?...
   ... Yes. C[1,2]:=C[0,1]+1
   ... C[0,1] is 0. C[1,2]:=1
<state>
i=1 j=2 M=2 N=3
C[0,0]=0 C[0,1]=0 C[0,2]=0 C[0,3]=0
C[1,0]=0 C[1,1]=0 C[1,2]=1 C[1,3]=0
</state>
j:=3
Check if a[1]==b[3]?  a[1] is T b[3] is A Is T==A?...
   ... No. C[1,3]:=detailed_max(C[1,2],C[0,3])
   ... C[1,2] is 1, C[0,3] is 0. C[1,3] is the greater of
   ...them. C[1,3]:=1
<state>
i=1 j=3 M=2 N=3
C[0,0]=0 C[0,1]=0 C[0,2]=0 C[0,3]=0
C[1,0]=0 C[1,1]=0 C[1,2]=1 C[1,3]=1
</state>
i:=2
j:=1
Check if a[2]==b[1]?  a[2] is A b[1] is A Is A==A?...
   ... Yes. C[2,1]:=C[1,0]+1
   ... C[1,0] is 0. C[2,1]:=1
<state>
i=2 j=1 M=2 N=3
C[0,0]=0 C[0,1]=0 C[0,2]=0 C[0,3]=0
C[1,0]=0 C[1,1]=0 C[1,2]=1 C[1,3]=1
C[2,0]=0 C[2,1]=1 C[2,2]=0 C[2,3]=0
</state>
j:=2
Check if a[2]==b[2]?  a[2] is A b[2] is T Is A==T?...
   ... No. C[2,2]:=detailed_max(C[2,1],C[1,2])
   ... C[2,1] is 1, C[1,2] is 1. C[2,2] is the greater of
   ...them. C[2,2]:=1
<state>
i=2 j=2 M=2 N=3
C[0,0]=0 C[0,1]=0 C[0,2]=0 C[0,3]=0
C[1,0]=0 C[1,1]=0 C[1,2]=1 C[1,3]=1
C[2,0]=0 C[2,1]=1 C[2,2]=1 C[2,3]=0
</state>
j:=3
Check if a[2]==b[3]?  a[2] is A b[3] is A Is A==A?...
   ... Yes. C[2,3]:=C[1,2]+1
   ... C[1,2] is 1. C[2,3]:=2
<state>
i=2 j=3 M=2 N=3
C[0,0]=0 C[0,1]=0 C[0,2]=0 C[0,3]=0
C[1,0]=0 C[1,1]=0 C[1,2]=1 C[1,3]=1
C[2,0]=0 C[2,1]=1 C[2,2]=1 C[2,3]=2
</state>
<state>
END
</state>
```

**Prompt A.4.** Full single path execution path prompt using the state description as in the fragmented Prompt 3. Playground link (use with 0 temperature)

```
Problem: 2, 3, 1, 5
EXECUTION
    Length of the list: L=4
    Number of pairs: P=3
    a=[2 3 1 5]
    set n_swaps=0. set i=P=3. set swap_flag=true.
        <state> a=[2 3 1 5] i=3 P=3 n_swaps=0 swap_flag=true </state>
    Since i=3 and P=3, i and P are equal, so this iteration is done, but swap_flag is true,
    so we need another iteration
    Iteration:
        set swap_flag=false.  set i=0. The state is:
        <state> a=[2 3 1 5] i=0 P=3 n_swaps=0 swap_flag=false </state>
        Since i=0 and P=3, these two are different, so we continue
        a[i]=a[0]=2 a[i+1]=a[1]=3
        Because 2<3 is true we keep state as is and move on by increasing i
        <state> a=[2 3 1 5] i=1 P=3 n_swaps=0 swap_flag=false </state>
        Since i=1 and P=3, these two are different, so we continue
        a[i]=a[1]=3 a[i+1]=a[2]=1
        Because 3<1 is false we set swap_flag=true,increase n_swaps by one, and in a=[2 3 1 5] swap 3 and 1,
        and increase i, and keep P as is to get
        <state> a=[2 1 3 5] i=2 P=3 n_swaps=1 swap_flag=true </state>
        Since i=2 and P=3, these two are different, so we continue
        a[i]=a[2]=3 a[i+1]=a[3]=5
        Because 3<5 is true we keep state as is and move on by increasing i
        <state> a=[2 1 3 5] i=3 P=3 n_swaps=1 swap_flag=true </state>
        Since i=3 and P=3, these two are equal, so this iteration is done, but swap_flag is true,
        so we need another iteration
    Iteration:
        set swap_flag=false.  set i=0. The state is:
        <state> a=[2 1 3 5] i=0 P=3 n_swaps=1 swap_flag=false </state>
        Since i=0 and P=3, these two are different, so we continue
        a[i]=a[0]=2 a[i+1]=a[1]=1
        Because 2<1 is false we set swap_flag=true,increase n_swaps by one, and in a=[2 1 3 5] swap 2 and 1,
        and increase i, and keep P as is to get
        <state> a=[1 2 3 5] i=1 P=3 n_swaps=2 swap_flag=true </state>
        Since i=1 and P=3, these two are different, so we continue
        a[i]=a[1]=2 a[i+1]=a[2]=3
        Because 2<3 is true we keep state as is and move on by increasing i
        <state> a=[1 2 3 5] i=2 P=3 n_swaps=2 swap_flag=true </state>
        Since i=2 and P=3, these two are different, so we continue
        a[i]=a[2]=3 a[i+1]=a[3]=5
        Because 3<5 is true we keep state as is and move on by increasing i
        <state> a=[1 2 3 5] i=3 P=3 n_swaps=2 swap_flag=true </state>
        Since i=3 and P=3, these two are equal, so this iteration is done, but swap_flag is true,
        so we need another iteration
    Iteration:
        set swap_flag=false.  set i=0. The state is:
        <state> a=[1 2 3 5] i=0 P=3 n_swaps=2 swap_flag=false </state>
        Since i=0 and P=3, these two are different, so we continue
        a[i]=a[0]=1 a[i+1]=a[1]=2
        Because 1<2 is true we keep state as is and move on by increasing i
        <state> a=[1 2 3 5] i=1 P=3 n_swaps=2 swap_flag=false </state>
        Since i=1 and P=3, these two are different, so we continue
        a[i]=a[1]=2 a[i+1]=a[2]=3
        Because 2<3 is true we keep state as is and move on by increasing i
        <state> a=[1 2 3 5] i=2 P=3 n_swaps=2 swap_flag=false </state>
        Since i=2 and P=3, these two are different, so we continue
        a[i]=a[2]=3 a[i+1]=a[3]=5
        Because 3<5 is true we keep state as is and move on by increasing i
        <state> a=[1 2 3 5] i=3 P=3 n_swaps=2 swap_flag=false </state>
        Since i=3 and P=3, these two are equal, so this iteration is done, but swap_flag is false, so we are done
    Final List: 1, 2, 3, 5
    Number of swaps: 2
    END OF EXECUTION

Problem: 3, 6, 8, 2, 7
EXECUTION
```

**Prompt A.5.** Prompt that triggers execution of the search for the longest substring without repeating characters. Playground link (use 0 temperature)

```
Input: s = c, b, c, a, b, b
START
Unique letters: a, b, c
Define variables last_a=0, last_b=0, last_c=0
Length of sequence s:  L=6
Because L is 6, the needed number of iterations is 6
set st_ind=1
st m_len=0
set i=1
Iteration 1:
    s(1) is c, so use last_c
    last_c is 0, so  nothing to do here.
    max(m_len,  i-st_ind+1) is max(0, 1-1+1) which is...
    ...max(0,1)=1, so we set m_len=1
    since i is 1, and the letter is c, set last_c=1
    increase i by one
    i=2, st_ind=1, m_len=1, last_a=0, last_b=0, last_c=1
End of iteration 1. But we need to do 6 iterations,...
...so we do another one
Iteration 2:
    s(2) is b, so use last_b
    last_b is 0, so  nothing to do here.
    max(m_len,  i-st_ind+1) is max(1, 2-1+1) which is...
    ...max(1, 2)=2, so we set m_len=2
    since i is 2, and the letter is b, set last_b=2
    increase i by one
    i=3, st_ind=1, m_len=2, last_a=0, last_b=2, last_c=1
End of iteration 2.  But we need to do 6 iterations,...
...so we do another one
Iteration 3:
    s(3) is c, so use last_c
    last_c is greater than 0, so we reason...
    ...max(st_ind, last_c+1) is max(1, 2)=2...
    ...so we set st_ind=2
    max(m_len,  i-st_ind+1) is max(2, 3-2+1) which is...
    ...max(2, 2)=2, so we set m_len=2
    since i is 3, and the letter s(3) is c, set last_c=3
    increase i by one
    i=4, st_ind=2, m_len=2, last_a=0, last_b=2, last_c=3
End of iteration 2. But we need to do 6 iterations,...
...so we do another one
Iteration 4:
    s(4) is a, so use last_a
    last_a is 0, so  nothing to do here.
    max(m_len,  i-st_ind+1) is max(2, 4-2+1) which is...
    ...max(2, 3)=3, so we set m_len=3
    since i is 4, and the letter s(4) is a, set last_a=4
    increase i by one
    i=5, st_ind=2, m_len=3, last_a=4, last_b=2, last_c=3
End of iteration 4. But we need to do 6 iterations,...
...so we do another one
Iteration 5:
    s(5) is b, so use last_b
    last_b is greater than 0, so we reason...
    ...max(st_ind, last_b+1) is max(2, 2+1) which is...
    ...max(2, 3)=3 so we set st_ind=3
    max(m_len,  i-st_ind+1) is max(3, 5-3+1) which is...
    ...max(3, 3)=3, so we set m_len=3
    since i is 5, and the letter s(5) is b, set last_b=5
    increase i by one
    i=6, st_ind=3, m_len=3, last_a=4, last_b=5, last_c=3
End of iteration 5. But we need to do 6 iterations,...
...so we do another one
Iteration 6:
    s(6) is b, so use last_b
    last_b is greater than 0, so we reason...
    ...max(st_ind, last_b+1) is max(3, 5+1) which is...
    ...max(3, 6)=6 so we set st_ind=6
    max(m_len,  i-st_ind+1) is max(3, 6-6+1) which is...
    ...max(3, 1)=3, so we set m_len=3
    since i is 6, and the letter s(6) is b, set last_b=6
    increase i by one
    i=7, st_ind=6, m_len=3, last_a=4, last_b=6, last_c=3
End of iteration 6. We needed to do 6 iterations,...
...so we are done

The solution is: m_len=3
END

Input: s = p, w, w, k, e, p, z
START
```

**Prompt A.6.** Prompt that triggers evaluation of parentheses using a stack. Full prompt in playground, and Prompt as here (with 10 steps missing) in playground. Meant to be used with 0 temperature.

```
input: ) [ { } ] ( { } ) [ ( { } ) ] } {
input wriritten as a sequence of symbols:
s= ')', '[', '{', '}', ']', '(', '{','}', ')', '[',
'(', '{', '}', ')',']', '}', '{'
length(s)= 17
stack is initialized as empty
i=0
there is nothing in stack, so push s(0)= ')' on stack
stack= )
are the last two symbols an open and a closed
parenthesis of the same type? No. Stack stays same.
i=1
we push s(1)='[' on the stack
stack= ) [
are the last two symbols an open and a closed
parenthesis of the same type? No. Stack stays same.
i=2
we push s(2)='{' to the stack
stack= ) [ {
are the last two symbols an open and a closed
parenthesis of the same type? No. Stack stays same.
i=3
we push s(3)='}' to the stack
stack= ) [ { }
are the last two symbols an open and a closed
parenthesis of the same type? Yes, they are { },
opening then closing.
We pop the last two symbols from the stack.
stack= ) [
i=4
we push s(4)=']' to the stack
stack= ) [ ]
are the last two symbols an open and a closed
parenthesis of the same type? Yes, they are [ ],
opening then closing.
We pop the last two symbols from the stack
stack= )
i=5
we push s(5)='(' to the stack
stack= ) (

...

i=15
we push s(15)='}' to the stack
stack= ) }
are the last two symbols an open and a closed
parenthesis of the same type? No. Stack stays same.
i=16
we push s(16)='{' to the stack
stack= ) } {
are the last two symbols an open and a closed
parenthesis of the same type? No. Stack stays same.
i=17
we have reached the end of the input string.
If the stack has some parenthesis left in it,
the sequence is invalid, otherwise,
if the stack is empty, it is valid.
Sequence is: invalid
END

input:
```

**Prompt A.7.** A prompt (white background) for translating logical deduction puzzles from BIG-bench into a Wolfram language (Mathematica) call. The response from GPT-3 Codex is shown green-shaded. Playground link (use 0 temperature)

```
Puzzle: The following objects need to be ordered. obj1 is the biggest. obj2 is smaller than obj3.
obj1 is bigger than obj2.

Question: Which object is the biggest?

Parsing step:
    Items: obj1, obj2, obj3
    Numbe of items: 3
    Statement 1: obj1 is the biggest.
    Statement 2: obj2 is smaller than obj3.
    Statement 3: obj1 is bigger than obj2.

Scoring identification step:
     Scores will refer to size.
     Since we have 3 items, let's assume that the biggest gets a score  of 3 pounds and the smallest gets
     the score of 1 pound.

Translation step:
    Available variable names: x, y, z, a, b, c
    Map item scores of 'obj1', 'obj2', 'obj3' to variable names x, y, z
    obj1 score is x; obj2 score is y; obj3 is z;
    Statement 1: 'x' is the biggest.
    Statement 2: 'y' is smaller than 'z'.
    Statement 3:  'x' is bigger than 'y'.
    Domain constraints: 1<=x<=3, 1<=y<=3, 1<=z<=3
    Statemets as a set of inequalities or equalities: x==3, y<z, x>y
    All varaibles are different: x!=y, x!=z, y!=z

Call to WolframAlpha:
    Solve[{1<=x<=3, 1<=y<=3, 1<=z<=3, x==3, y<z, x>y, x!=y, x!=z, y!=z }, {x, y, z}, Integers]

END

Puzzle: A fruit stand sells five fruits: kiwis, pears, peaches, loquats, and apples.
The loquats are more expensive than the kiwis. The pears are more expensive than the peaches.
The apples are more expensive than the loquats. The apples are less expensive than the peaches.

Question: Which is the most expensive?
```

```
Parsing step:
    Items: kiwis, pears, peaches, loquats, apples
    Numbe of items: 5
    Statement 1: The loquats are more expensive than the kiwis.
    Statement 2: The pears are more expensive than the peaches.
    Statement 3: The apples are more expensive than the loquats.
    Statement 4: The apples are less expensive than the peaches.

Scoring identification step:
     Scores will refer to price.
     Since we have 5 items, let's assume that the most expensive gets a score  of 5 dollars
     and the least expensive gets the score of 1 dollar.

Translation step:
    Available variable names: x, y, z, a, b, c
    Map item scores of 'kiwis', 'pears', 'peaches', 'loquats', 'apples' to variable names x, y, z, a, b
    kiwis score is x; pears score is y; peaches is z; loquats is a; apples is b;
    Statement 1: 'a' is more expensive than 'x'.
    Statement 2: 'y' is more expensive than 'z'.
    Statement 3:  'b' is more expensive than 'a'.
    Statement 4:  'b' is less expensive than 'z'.
    Domain constraints: 1<=x<=5, 1<=y<=5, 1<=z<=5, 1<=a<=5, 1<=b<=5
    Statemets as a set of inequalities or equalities: a>x, y>z, b>a, b<z
    All varaibles are different: x!=y, x!=z, x!=a, x!=b, y!=z, y!=a, y!=b, z!=a, z!=b, a!=b

Call to WolframAlpha:
    Solve[{1<=x<=5, 1<=y<=5, 1<=z<=5, 1<=a<=5, 1<=b<=5, a>x, y>z, b>a, b<z, x!=y, x!=z, x!=a, x!=b,
    y!=z, y!=a, y!=b, z!=a, z!=b, a!=b}, {x, y, z, a, b}, Integers]
```

**Prompt A.8.** A zero-shot code prompt (white background) which, when temperature is carefully chosen may trigger code execution (shaded green) by Codex. Although the sequence gets sorted, the execution goes awry after that. Importantly, the number of swaps is not tracked, even though the code contains appropriate instructions. In contrast, IRSA prompts consistently and repeatedly trigger correct program execution. See also Prompts A.11 and A.12 for examples of inconsistent behaviour of GPT-4 with zero-shot code prompting.

```
# Here is the code for Bubble Sort algorithm that keeps track of the number of needed swaps:
def bubbleSort(arr):
    n = len(arr)
    # initiate the number of swaps to 0
    n_swaps=0;
    # Traverse through all array elements
    for i in range(n):

        # Last i elements are already in place
        for j in range(0, n-i-1):

            # traverse the array from 0 to n-i-1
            # Swap if the element found is greater
            # than the next element
            if arr[j] > arr[j+1] :
                arr[j], arr[j+1] = arr[j+1], arr[j]
                n_swaps = n_swaps+1

Problem: 8, 7, 6, 4, 9
EXECUTION:
```

```
    i=0
    j=0
    8 > 7
    7, 8, 6, 4, 9
    j=1
    8 > 6
    7, 6, 8, 4, 9
    j=2
    8 > 4
    7, 6, 4, 8, 9
    j=3
    8 > 9
    7, 6, 4, 8, 9
    j=4
    i=1
    j=0
    7 > 6
    6, 7, 4, 8, 9
    j=1
    7 > 4
    6, 4, 7, 8, 9
    j=2
    7 > 8
    6, 4, 7, 8, 9
    j=3
    i=2
    j=0
    6 > 4
    4, 6, 7, 8, 9
    j=1
    6 > 7
    4, 6, 7, 8, 9
    j=2
    i=3
    j=0
    4 > 6
    4, 6, 7, 8, 9
    j=1
    i=4
    j=0
    i=5
    j=0
    i=6
    j=0
    i=7
    j=0
    i=8
    ...
```

**Prompt A.9.** An example of a baseline prompt design. Prompts with and without the code, shaded in gray, were used. Zero shot prompts omitted the examples, also shaded in gray. Other than the (buggy) algorithm we adopted for the logical deduction puzzle, the algorithms we tested are known to Codex, and we based the codes used in the prompts on the ones generated by Codex itself.

```
# Python3 program to find the length
# of the longest substring
# without repeating characters in string s
# the maximum length of such a substring will be returned in m_len
```

```
def longestUniqueSubsttr(s):
    # last index of every character
    last_idx = {}
    m_len = 0
    # starting index of current
    # window to calculate m_len
    start_idx = 0
    for i in range(0, len(s)):
        # Find the last index of str[i]
        # Update start_idx (starting index of current window)
        # as maximum of current value of start_idx and last
        # index plus 1
        if s[i] in last_idx:
            start_idx = max(start_idx, last_idx[s[i]] + 1)

        # Update result if we get a larger window
        m_len = max(m_len, i-start_idx + 1)
        # Update last index of current char.
        last_idx[s[i]] = i
    return m_len
```

```
What would the algorithm above, which looks for the longest substring with non repeating characters
compute for a given problem? Use this format:

Input: s = ...
START
The solution is: m_len= ...
END
```

```
Input: s = a, a, c, a, d, b, b
START
The solution is: m_len=4
END

Input: s = b, b, a, a, d, b, b
START
The solution is: m_len=3
END

Input: s = c, a, d, c, b, c, a
START
The solution is: m_len=4
END

Input: s = a, c, b, a, c, d, d
START
The solution is: m_len=4
END

Input: s = b, b, d, d, a, d, b
START
The solution is: m_len=3
END
```

```
Input: s = a, c, d, c, a, c, a
START
```

**Prompt A.10.** Given the non-shaded text as a prompt, GPT-4 (0 temperature) returns the green shaded response.

```
We need to compute the longest common subsequence for two sequences

s1=bccba
s2=ccaa

using the dynamic programming algorithm. Show the python code for the algorithm, and then execute it.
Finally, output the length of the longest common subsequence bracketed with <answer> and </answer>.
```

Here's the Python code for the dynamic programming algorithm to find the longest common subsequence:

```python
def longest_common_subsequence(s1, s2):
    m = len(s1)
    n = len(s2)
    dp = [[0] * (n + 1) for _ in range(m + 1)]

    for i in range(1, m + 1):
        for j in range(1, n + 1):
            if s1[i - 1] == s2[j - 1]:
                dp[i][j] = dp[i - 1][j - 1] + 1
            else:
                dp[i][j] = max(dp[i - 1][j], dp[i][j - 1])

    return dp[m][n]

s1 = "bccba"
s2 = "ccaa"
lcs_length = longest_common_subsequence(s1, s2)
print(f"<answer>{lcs_length}</answer>")
```

Now, let us execute the code and find the length of the longest common subsequence:

```python
s1 = "bccba"
s2 = "ccaa"
lcs_length = longest_common_subsequence(s1, s2)
print(f"<answer>{lcs_length}</answer>")
```

Output:

```
<answer>3</answer>
```

**Prompt A.11.** Given the non-shaded text as a prompt, GPT-4 (0 temperature) returns the green shaded response. The prompt attempts to trigger tracking steps of program execution, but only the initial and final matrix are shown.

```
We need to compute the longest common subsequence for two sequences

s1=bccba
s2=ccaa

using the dynamic programming algorithm. Show the python code for the algorithm, and then write down its
execution with intermediate steps. Finally, output the length of the longest common subsequence bracketed
with <answer> and </answer>.
```

Here's the Python code for the dynamic programming algorithm to find the longest common subsequence:

```python
def longest_common_subsequence(s1, s2):
    m = len(s1)
    n = len(s2)
    dp = [[0] * (n + 1) for _ in range(m + 1)]

    for i in range(1, m + 1):
        for j in range(1, n + 1):
            if s1[i - 1] == s2[j - 1]:
                dp[i][j] = dp[i - 1][j - 1] + 1
            else:
                dp[i][j] = max(dp[i - 1][j], dp[i][j - 1])

    return dp[m][n]

s1 = "bccba"
s2 = "ccaa"
print(longest_common_subsequence(s1, s2))
```

Now let us s go through the execution with intermediate steps:

1. Initialize the 'dp' table with dimensions (m+1) x (n+1), where m and n are the lengths of s1 and s2, respectively. The table will look like this:

```
0 0 0 0 0
0 0 0 0 0
0 0 0 0 0
0 0 0 0 0
0 0 0 0 0
0 0 0 0 0
```

2. Iterate through the table, comparing characters from s1 and s2. Update the table based on the dynamic programming formula:

```
0 0 0 0 0
0 0 1 1 1
0 1 1 1 1
0 1 2 2 2
0 1 2 2 2
0 1 2 2 3
```

3. The length of the longest common subsequence is in the bottom-right cell of the table, which is 3.

So, the length of the longest common subsequence is <answer>3</answer>.

**Prompt A.12.** Given the same non-shaded text as in Prompt A.11 as a prompt, **just for a different input**, GPT-4 (0 temperature) returns the green shaded response that this time does list more (but not all!) intermediate steps. This style of prompting GPT-4 reaches accuracy of 69% on LCS-S, compared to 93% achieved with IRSA in GPT-3.

```
We need to compute the longest common subsequence for two sequences

s1=aaca
s2=abab

using the dynamic programming algorithm. Show the python code for the algorithm, and then write down its
execution with intermediate steps. Finally, output the length of the longest common subsequence bracketed
with <answer> and </answer>.
```

Here is the Python code for the dynamic programming algorithm to compute the longest common subsequence:
```python
def longest_common_subsequence(s1, s2):
    m = len(s1)
    n = len(s2)

    dp = [[0]*(n+1) for _ in range(m+1)]

    for i in range(m+1):
        for j in range(n+1):
            if i == 0 or j == 0:
                dp[i][j] = 0
            elif s1[i-1] == s2[j-1]:
                dp[i][j] = dp[i-1][j-1] + 1
            else:
                dp[i][j] = max(dp[i-1][j], dp[i][j-1])

    return dp[m][n]

s1 = "aaca"
s2 = "abab"
result = longest_common_subsequence(s1, s2)
print(result)
```
Here is the execution with intermediate steps:

1. Initialize the 'dp' table with dimensions (m+1) x (n+1). For the given sequences, it will look like:

```
   0 0 0 0 0
0  0 0 0 0 0
0  0 0 0 0 0
0  0 0 0 0 0
0  0 0 0 0 0
```

2. Fill in the 'dp' table by iterating through each cell and apply the algorithm rules:

For i, j = 1, 1: s1[i-1] != s2[j-1], so dp[i][j] = max(dp[i-1][j], dp[i][j-1]) = max(0, 0) = 0
```
   0 0 0 0 0
0  0 0 0 0 0
0  0 0 0 0 0
0  0 0 0 0 0
0  0 0 0 0 0
```

For i, j = 1, 2: s1[i-1] != s2[j-1], so dp[i][j] = max(dp[i-1][j], dp[i][j-1]) = max(0, 0) = 0
```
   0 0 0 0 0
0  0 0 0 0 0
0  0 0 0 0 0
0  0 0 0 0 0
0  0 0 0 0 0
```

For i, j = 1, 3: s1[i-1] == s2[j-1], so dp[i][j] = dp[i-1][j-1] + 1 = 0 + 1 = 1
```
   0 0 0 0 0
0  0 0 0 0 0
0  0 0 1 0 0
0  0 0 0 0 0
0  0 0 0 0 0
```

Continue filling in the table with the same rules:

```
   0 0 0 0 0
0  0 0 0 0 0
0  0 0 1 0 0
0  0 1 1 1 0
0  0 1 1 2 0
```

3. The value in the bottom-right cell contains the length of the longest common subsequence:

<answer>2</answer>