# OpenReview forum: "GPT is becoming a Turing machine: Here are some ways to program it"
_NeurIPS.cc/2023/Conference — Submitted to NeurIPS 2023_

### Official Review · Reviewer_tn6w · 2023-07-04

**Soundness:** 3 good
**Presentation:** 2 fair
**Contribution:** 1 poor
**Rating:** 4
**Confidence:** 4

**Summary:**

Authors propose a way to prompt GPT-3 to exhibit behavior simulating execution of iterative programs. Authors propose the following prompt constructs: providing structured examples of program execution; using fragments of execution; not using self-attention on some parts of the generated text. Authors compare the results to baselines and show significant improvements

**Strengths:**

- Authors introduce a method that enables GPT-3 to mimic the execution of iterative programs. They achieve this by supplying the model with intermediate steps and outcomes.
This is somewhat novel and could be useful for using LLMs to solve problems that require iterative processing.
- The use of path fragments may prove beneficial in situations where the context size is insufficient for comprehensive examples.
- The strategy of confining self-attention to particular segments of the output might be advantageous when the context size needs to be considered
- The examples provided in the paper are well written

**Weaknesses:**

- Authors' approach requires the manual construction of prompts for each problem at hand. It is not automated and not scalable. This limits the usefulness of the approach in practice.
- Authors compare their approach to simple baselines. There should be a comparison to at least chain-of-thought reasoning.
- Paper is hard to read and accept as a standalone without appendices. Authors refer to the content in the appendices too much.
- The significance of the work is low. It is known that LLMs can produce iterative output. Although the authors have enhanced the quality of such outputs via structured prompting, structured prompting is not entirely novel. Same can be said about using fragments in prompt/context.

EDIT: I have raised my evaluation of the paper from 3 to 4. Authors have promised to address the issues I and other reviewers have raised. However, in my opinion, such changes would require a major rewrite of the paper. I am not confident if these changes can be done well for the publication.

I have read the author’s rebuttal and further discussion with authors based on my questions and feedback.
Authors' rebuttal addressed some of my concerns by more in depth discussion of IRSA relationship to chain-of-thought reasoning and possible automated generation of IRSA prompts.

**Questions:**

Is there a generic prompt structure that could be used for any algorithm or some class of algorithms?

In Prompt 3 is the “Final List: 6, 7, 3, 5” intentional? It is not sorted.

**Limitations:**

It would be good if authors explored the limitations of what can be achieved by their approach. At some point the LLM "execution" (simulation really) of the program should fail. The points at which the simulation would fail likely depend on the input/output/context size. It may also depend on the algorithm semantic and/or algorithmic complexity. (Those are different complexities and may affect the breaking point differently). These questions could be explored and would be useful to know.

---

> ### Author Rebuttal · Authors · 2023-08-10
>
> For any program, we prompt the LLM with an execution path described well, but using arbitrary language (keywords and such). Therefore, an automatic way of creating IRSA prompts would be to simply generate an execution path programmatically, but following the rules of general repetitive structure, and providing inline explanations on each state transformation, as well as, importantly, which conditions to check to understand if a new iteration is necessary. Such well explained execution paths then serve as instructions to GPT how to execute the algorithm on new input (Prompt 1, a single execution for Bubble Sort, is an example of such a “program for GPT” that tells it to how to execute it for new inputs). So, our main point is that GPT can simulate execution of programs when prompted like this, and that fragmented prompting and skip attention further assist in this. This is important in two ways: 1. Executing algorithms by LLMs has been of interest to the community (as evidenced by various benchmarks), and it turns out this is quite doable using the ideas from our paper. 2. Complex reasoning tasks often masquerade as word problems when they can be solved with known algorithms if translated into a form consumable by those algorithms, and we show that LLMs can both perform that translation and run those algorithms (Logical deduction task) which is important for evaluation of such tasks with CoT prompts. (See also the general response on this.)
>
> Baseline CoTs: CoTs can vary dramatically, so in our experiments we focused on the ones that rely on program specification to reason about the input. In our experiments, we compared with few-shot and zero shot prompts with and without such programmatic instructions. Furthermore, in our GPT-4 comparisons, we induced execution path generation by GPT-4 with prompt A.11: GPT-4 shows its chain of thought reasoning on how the program should be executed. Still, it fails to match our results with IRSA. The reason for that is seen in the differences between A.11 and A.12 where the same prompting on different inputs induces different behaviors. With IRSA the behavior is much more consistent. As a result, GPT-3 with IRSA achieves 93% and GPT-4 with prompting in A.11, which can also execute a sort of state evolution and tracking, only reaches 63%. That is to say: GPT models can dazzle us with correct program execution sometimes, but IRSA raises the frequency of correct execution dramatically.
>
> Re Prompt 3, it is an example of a fragmented prompt, where each fragment starts with a state and ends with an explained transformation of that state. Some of these starting states are impossible, like the state just before “Final List: 6, 7, 3, 5”. However, the description of how such a state would be processed if reached is accurate. Correct transformations are what is needed for correct program execution, even if they are illustrated on unreachable states. When these transformations are executed correctly starting from the beginning of the execution, the incorrect states are never (or rarely) reached (as seen in experiments).
>
> Regarding a generic prompt structure, one can decide to always use the same set of keywords to describe state transformations. Furthermore, as we show in Prompt A.2 and A.3 in the Appendix, we can even create a prompt that will compile execution paths in a consistent language given a program description. Choice of language is up to the user, but nothing prevents them from using an existing one. We have an example of this with the dynamic programming prompt (2.4) which was generated using the strategy highlighted in Prompts A.2 and A.3. Using this prompt, other “programs for GPT” can be created following the same syntax.

---

> > ### Comment · Reviewer_tn6w · 2023-08-12
> > **Thanks to the authors for their response**
> >
> > I appreciate the comments by the authors with responses to my questions.
> >
> > In my opinion a general and automated way of creating IRSA prompts would increase the contribution a lot. It is worth not just a comment in the response, but rather actual implementation and evaluation - hopefully in the future work or future versions of this paper.
> > I don't believe that automation is as simple as authors claim. In fact, the authors themselves show how complex it is to create IRSA prompts for problems: someone has to write an algorithm - even better a program - then pick the important state of the program and construct it all into the IRSA prompt. This might be automatable via some program interpreter, but it is not obviously trivial.
> >
> > Based on authors' comments, it seems that IRSA is really one approach to specify a very structured CoT context (prompts) to the LLMs. I think this is a key way to look at what authors have done and how to evaluate their work. As authors correctly observe, the CoT is well known, widely adopted and practiced, and also can vary dramatically. This raises a difficulty of comparing the authors' approach to "CoT baseline(s)" since - as authors observe - such baselines can vary a lot both in content and in results. It would be worthwhile to try to specify some distribution of CoT baselines, e.g. prompts that just ask LLM to use CoT; prompts that provide input-output example; prompts that provide an execution example (which gets close to IRSA); and possibly others. In the future IRSA could be considered as one specific CoT approach to be compared against. However, this emphasizes even more the need to automate the way of creating IRSA prompts, since the issue authors raise regarding CoT prompts ("vary dramatically") also is present in IRSA prompts if they are manually written for each problem and algorithm.
> >
> > I remain concerned that the paper does not present the main important material in self-contained manner. The number of references to Appendixes is numerous (there are 33 references to Appendixes in the paper!). A lot of responses of authors also point to Appendixes. Appendixes themselves are not well written or readable, even if readers could access them. They understandably look like paragraphs that did not fit the paper and were thrown into Appendixes. I think the readers should be able to read a paper that is clear and self-contained.

---

> > > ### Author Response · Authors · 2023-08-18
> > > **Thanks for reviewer tn6w's comments and insightful discussion (1)**
> > >
> > > We thank a lot for Reviewer tn6w's great comments and engagement in the discussion (esp. with other reviewers)! We try to address each of the remaining concerns as follows:
> > >
> > >
> > > (1-a) **Automation of IRSA prompts**. We appreciate the reviewer's perspective on the importance of automation. To address this concern, we'd like to emphasize the following points.
> > >
> > > - First, as reviewer AAhn pointed out, the topic of the paper is algorithm execution, not discovery, which is in itself an unsolved problem, except for the fact that many (if not most) practical problems rely on a small number of known core algorithms.
> > > - Second, similarly, automated prompting is a very active research area now and is not a solved problem. In fact, it is only starting to be studied by the community and it's usually invented after some novel manual construction of those prompts. For example, the original CoT prompting requires manually writing down few-shot examples per task to trigger the reasoning process, sharing a similar manual construction process as ours. Therefore, we agree with the reviewer that automating IRSA prompting could be a promising research direction to explore in the future. Meanwhile, as emphasized in our general response (1) and our first paragraph in the first response to reviewer tn6w, our major contributions in this paper are slightly different from proposing an automatic way of constructing IRSA prompts.
> > > - Third, regarding the difficulty of automation, we recommend our second bubble sort prompt (Prompt 3 in the main text, also see the full basic ISRA in Appendix Prompt A.4) as a template for achieving the best results. This prompt structure encapsulates the benefits and main design of IRSA prompting, offering a strong reference for the potential automation process of creating IRSA prompts. Moreover, somehow related to the reviewer AAHn's comment on this point (see the last paragraph in https://openreview.net/forum?id=ARJG1kr8A7&noteId=M2edKRYFRp), when the algorithm to solve the problem is given, IRSA prompts could be crafted relatively easier. However, we agree with the reviewers that finding the correct algorithm solution is an important future direction to explore following up our work.
> > >
> > >
> > > Following the reviewer's suggestion, we will add a thorough discussion and implementation details to such an important issue. Furthermore, the results from fragmented prompting on bubble sort underscore our point: despite variations in the prompt (any fragments to use), the performance remains consistently high. This indicates that our IRSA approach has a level of robustness and the choice of program pieces is not an obstacle for the automation process. We hope our response could resolve the reviewer's concern.
> > >
> > >
> > >
> > > (1-b) **Comparison with CoT**. We appreciate the reviewer's suggestion of clarifying the relationship with CoT and a more systematic evaluation of various CoT baselines.
> > > - First, please refer to our response to reviewer n7ti for a complete response to clarify the definition of IRSA and its relationship with CoT.
> > > - Second, regarding the infinite variation of what CoT recipes may include, we agree with you that this makes CoT comparisons difficult. Part of the message of the paper is that IRSA, as a special kind of CoT, as all reviewers agree, leads to a surprising level of accuracy (but not perfection) in the execution of algorithms.
> > > - Third, among all the baselines the reviewer suggested, we currently have compared with one of the most competitive ones, prompts asking for CoT using GPT-4 (prompt A.11 and A.12), which gets 69\% accuracy on LCS-Short, while our ISRA with skip attention gets 93\% on CodeX. We believe such a close comparison clearly demonstrates the benefits of our IRSA prompt design. To address this concern, we will move such results from the appendix to the main text and clearly explain them in the main experiment section.

---

> > > > ### Author Response · Authors · 2023-08-18
> > > > **Thanks for reviewer tn6w's comments and insightful discussion (2)**
> > > >
> > > > (1-c\) **Paper organization and dependence on appendix**. We understand the reviewer's concern regarding the frequent references to the appendix.
> > > > - First, we believe showing the full prompt for each example is super important to understand our design, also pointed out by the reviewer, `The examples provided in the paper are well written`. Thus, many interesting prompts are moved to the appendix due to their long length (usually one prompt per page) and the limited space for the submission. However, our main text has already included the basic IRSA prompt, as well as the fragmented and skip attention prompts where the audiences could understand them directly based on the main text.
> > > > - Second, we assure the reviewer that this is an aspect we can rectify easily (as specified in the global response (2)), especially with the allowance of an extra page. Our primary goal is to enhance the clarity and coherence of the paper, ensuring that the reader can grasp the core ideas without excessive cross-referencing. We hope such a commitment could help resolve the concern of the reviewer on the presentation without interfering with the judgment of the contributions of our work.
> > > >
> > > >
> > > > Finally, we are grateful for your comments on other reviewers' responses and we've tried our best to summarize and address the main points all in the above responses, which should further clarify our major contributions to the field (let’s also not forget the fragmented prompting and skip attention). Given the consensus among the majority of the reviewers, we hope our work has brought something that the research community should be aware of, ensuring we don't throw the baby out with the bathwater, so-to-speak. It would also be our honor if our responses may prompt a reconsideration of the paper's rating. And if there are any other questions, please let us know.

---

> > > > ### Comment · Reviewer_tn6w · 2023-08-19
> > > > **Thanks to the authors for further elaboration and answers**
> > > >
> > > > I appreciate the answers of the authors and their elaboration.
> > > >
> > > > I wanted to briefly comment on one point.
> > > > Authors say: "many (if not most) practical problems rely on a small number of known core algorithms."
> > > >
> > > > I believe this misdirects from the complexities of programming large real-world systems.
> > > > Yes, everything can be implemented with arithmetic operators, branches, and goto. That's why assembly language works.
> > > > Possibly most programming relies on small number of known core algorithms.
> > > > In practice, real-world systems have intricate and complicated data structures with numerous inter dependencies.
> > > > The "algorithm" of doing something might be "known core algorithm" (more likely a number of such), but the state of the program is usually enormous. State changes may be very simple in each programming language step, but a large number of such steps is required to get from one internally consistent program state to another.
> > > > So the fact that algorithm might be simple does not make the whole system behavior simple or easy to infer from its execution path.
> > > >
> > > > This relates to the current paper: Even though providing execution paths for non-trivial algorithms with small states is possible, providing such paths for large real-world systems would require either enormous contexts or non-trivial pruning. Likely both.
> > > >
> > > > Applying IRSA to programs with large state spaces is not a task of this paper, so my comment aims to address just the authors' comment.

---

> > > > > ### Author Response · Authors · 2023-08-19
> > > > > **Fragmented prompting…**
> > > > >
> > > > > (Note that fragmented prompting, which was appreciated by the reviewer, provides a way to shorten the IRSA prompts and/or cover more examples/special cases; skip attention is also helpful. Looks like a lot of discussion was about basic IRSA…)

---

> > > > > > ### Comment · Reviewer_tn6w · 2023-08-21
> > > > > > **Raised the score from 3 to 4**
> > > > > >
> > > > > > I appreciate all the discussion from the authors. I have raised my evaluation of the paper from 3 to 4. Authors have promised to address the issues I and other reviewers have raised. However, in my opinion, such changes would require a major rewrite of the paper. I am not confident if these changes can be done well for the publication.

---

### Official Review · Reviewer_AAHn · 2023-07-06

**Soundness:** 2 fair
**Presentation:** 3 good
**Contribution:** 3 good
**Rating:** 6
**Confidence:** 3

**Summary:**

Making LLMs follow procedural rules precisely, as done when executing a program, is been a challenging task. In this paper, the authors introduce iterations by regimenting self-attention (IRSA), a prompting technique to make large-language models (LLMs) *execute* a hand coded programs (on novel inputs) precisely. The authors propose three techniques for IRSA: 1) by prompting the LLM with a step-by-step example showing many state-transitions in details, 2) By prompting with fragments of the state-to-state transitions only (along with the latest state for which it predicts the transition), and finally 3)skipping attention on intermediate state-to-state transitions. LLMs are shown to be significantly more successful at tasks such as sorting arrays, finding longest sub-sequence in a string, or simpler logical puzzles etc. when prompted with IRSA.

**Strengths:**

### Originality

Recently, many new proposals for prompting LLMs, including scratchpad, and Chain-of-Though prompting have been proposed. However, most approaches focus on making LLMs reason and solve problems/puzzles. Instead, in this paper, the focus is on making LLMs *follow instructions accurately* (which then can be used for solving certain puzzle). This is a novel and original direction.

### Quality
The paper is well presented, including the figures, and the tables. Additionally, the experiments have been conducted on many tasks to sho

### Clarity
The paper is very clearly written and well presented. Specifically, I found the prompt examples very useful in understanding the paper's ideas.

### significance
Getting LLMs to precisely execute procedural rules is of intereset to the research community at large. LLMs inability to successfully tackle procedural problems (such as multiplication) of complexity beyond the training set complexity has raised questions regarding its ability tackle compositional problems. This paper provide an important perspective and countering result that will further enrich this discussion.


**Weaknesses:**

The two main drawbacks of the paper are motivation and experiments.

### Motivation
The paper does not sufficiently motivate the problem statement. Why should we care about making LLMs execute programs - perform iterative behavior? When the process is deterministic and easy to programmatically describe, why would we prefer LLMs over a deterministic typical program (one can even use programs which explicitly *show* transition rules applied as well)? The authors mention education or software engineering vaguely, but there is no concrete motivation in these use-cases (when would a LLM be more suitable for this task over a REPL-like loop with python/cpp?).


### Experiments

1) The authors do not evaluate on significantly larger sequences sizes. Since fragmented prompting seems to allow arbitrarily large sequence of state-transitions, I believe authors can indeed use IRSA on larger sequence problems. The trend between success rate and sequence length would be insightful.
2) I strongly appreciate the authors for showing the negative result in Figure A.1 (Appendix section A.3.2). This shows that despite using IRSA, the model may end up performing wrong state-transitions based on its correlation to patterns in recent history. If multiple previous state transitions contain sequences where the statement "2 < x = True" appears, then when asked "2 < 1 =" the LLM has higher likelihood of filling True than False. This seems to directly negate the claim of this paper that we can make LLMs execute programs precisely. It clearly seems to be affected by the prompt history which can make them act in unreliable ways.

**Questions:**

I would appreciate if the authors can respond to the weaknesses raised above.

**Limitations:**

Yes the authors have adequately addressed the limitations and potential negative societal impact of their work.

---

> ### Author Rebuttal · Authors · 2023-08-10
>
> The versions of GPT we had available at the time of submission had 4k token limit. The problem with LCS is that its state is proportional to the product of sequence lengths (the DP matrix). With newer models, this will indeed be less of an issue (see the response to the previous reviewer, too). Nevertheless, the results on longer LCS problems from Big Bench still significantly beat the SOTA, and when we get a chance to run the experiments with APIs allowing higher token limits, we expect to do even better.
>
> We do not seek to recommend LLMs over deterministic programs for the execution of these programs, nor do we claim that LLMs can always execute programs precisely. Rather, we demonstrate the power of these models to perform the iterative steps necessary to execute an algorithm with high probability (empirically) on various algorithms. We describe the techniques required to trigger that sort of reasoning from the models as well as their limitations, such as with incorrectly correlating patterns and with solving larger problems. We explore how close we can get to accurate execution with prompting techniques that take advantage of existing architecture.
>
> See also the general response, on this question; briefly there is direct interest in ability of LLMs to execute programs in the research community. But, also, there is an indirect interest among those who study complex reasoning tasks that can be solved algorithmically: We’d like them to recognize that LLMs do not have to entangle the language understanding, common sense reasoning and algorithmic thinking in some cryptic form; instead prompts like ours for Logical Deduction can separate the processing of the word problem into an input to an algorithm from algorithm execution (by the LLM), making them equivalent to a combination of problem translation into a machine-readable form and a separate call to an algorithm (ran on a computer).
>
> And people who make the benchmarks should understand that it is possible to make CoT prompts that will trigger the kind of algorithmic reasoning they are interested in investigating regarding LLMs. Currently, one direction being used to evaluate/improve LLMs on complex reasoning tasks is to use benchmarks that are solvable algorithmically. We have shown that LLMs are in fact already capable of doing this through prompting that triggers iterative execution.

---

> > ### Comment · Reviewer_AAHn · 2023-08-16
> > **Thank you for the rebuttal!**
> >
> > ## Summary
> >
> > On one hand, the key contribution of the paper, a prompting strategy to make LLMs execute algorithms in an iterative step-by-step fashion, and the superior results to previous prompting strategies makes the work significant, and of high interest to the community. On the other hand, the paper has some drawbacks, namely 1) The presentation 2) lack of certain baselines and experiments.
> >
> > I am overall tending towards increasing my rating for the paper. The paper has its drawbacks, and its still unclear to me *when* this approach can be employed. However, it offers the insightful that LLMs can be (approximately) made to execute algorithms by simply changing the prompting strategy and this insight will likely intrigue researchers using LLMs at large.
> >
> >
> > ## Detailed response to the rebuttal
> >
> > >  do we claim that LLMs can always execute programs precisely.
> >
> > Thank you for correcting me. While the paper does not state that the LLM can execute programs precisely, that is the intended goal of the method (to make the LLM follow an algorithm *precisely*). My critique is that the proposed method seems highly susceptible to patterns in local history, making it more unreliable (i.e. IRSA might not work for executing algorithms which might have higher presence of recurrent patterns in its state-transition sequence).
> >
> > > perform the iterative steps necessary to execute an algorithm with high probability (empirically) on various algorithms
> >
> > I would appreciate if the authors can specify a simple clear use-case where we would like an LLM to perform iterative algorithmic steps. I believe a stronger (and more practical) alternative would be a a LLM-in-the-loop system where such programmatic iterative steps are handled by an existing coding language like python, and parts of the algorithm which involve uncertain semantics such as language understanding/common sense reasoning are performed via API-calls to the LLM (some simpler examples of such as are ViperGPT/VisProg). I think providing a concrete use-case will ground the reader's motivation.
> >
> > > See also the general ... (ran on a computer).
> >
> > I mostly agree.
> >
> > > prompts like ours for Logical Deduction can separate the processing of the word problem into an input to an algorithm from algorithm execution (by the LLM), making them equivalent to a combination of problem translation into a machine-readable form and a separate call to an algorithm (ran on a computer).
> >
> > In my understanding this decoupling of logical deduction into problem translation and separate call to an algorithm can already be achieved by using LLMs with code-interpreters. (i.e. asking the LLM to synthesize an algorithm in a programming language such as python, along with its inputs, and explicitly using the language's interpreter to run the algorithm). Do we even need IRSA for this decoupling?
> >
> > > We have shown that LLMs are in fact already capable of doing this through prompting that triggers iterative execution.
> >
> > With IRSA style prompts, we are only testing the LLM's ability to 1) translate natural language to create inputs to an algorithm, and 2) running the algorithm step-by-step. However, the benchmarks test is also 3) Selecting the right algorithm or creating the right algorithm for the problem. I am not sure (3) is tested in the current setup as the in-context examples use the same algorithm that we want the LLM to follow.
> >
> > ## Regarding Reviewer tn6w comments
> >
> > I agree the reviewer's critique regarding the paper's readability. In my opinion, the presence of full-page prompts in the main paper reduces the space available for exposition and experiments, pushing the material into the appendix. I believe if the revised versions reduce the space taken by the prompts, the authors may be able to pull back some of the material from appendix to the main draft.
> >
> >  I also agree that in some sense IRSA seems like "extremely structured" CoT prompting, and that presence of various gradations in types of CoT prompting in ablations would strongly improve the work.
> >
> > I however disagree with the reviewer on the difficulty of creating an algorithm's corresponding IRSA prompt. If an algorithm can be specified in pseudo-code, it is reasonable that a (python) program can be written to 1) consume the pseudo-code and convert to executable code, and then 2) follow the algorithm, and print all state variables at each step for some example inputs. Furthermore, the state can be pruned to show only relevant variables (which change over the course of the algorithm's execution etc.). I believe the key difficulty lies in crafting the algorithm itself, which I think this work does not address (and does not target addressing).

---

> > > ### Comment · Reviewer_tn6w · 2023-08-17
> > > **Thanks for reviewer AAHn comments regarding creation of IRSA prompts**
> > >
> > > I agree that the biggest hurdle for creating IRSA prompts is "crafting the algorithm itself" and that this paper does not and should not address algorithm creation.
> > >
> > > I partially disagree that automatic state pruning is simple or solved problem. If it was, we'd have debuggers showing abstract/relevant pruned state only during program execution, which is not the case mostly.
> > >
> > > Maybe it is enough to describe the IRSA prompt creation at the level of detail reviewer AAHn did without providing full implementation of their generation from algorithm/program.
> > >
> > > As I said, automatically generated IRSA prompts would facilitate achieving consistency, repeatability, and reproducibility of the IRSA prompting, so we would not have the variability that impacts prompts overall and CoT prompts too.
> > >
> > > Thanks

---

> > > > ### Comment · Reviewer_AAHn · 2023-08-18
> > > > **Thank you reviewer tn6w for the interesting counter-point!**
> > > >
> > > > Thank you Reviewer tn6w, for your thoughtful response.
> > > >
> > > > > I partially disagree that automatic state pruning is simple or solved problem. If it was, we'd have debuggers showing abstract/relevant pruned state only during program execution, which is not the case mostly.
> > > >
> > > > That is an interesting counter-point! Un-pruned state-variables can indeed be a source of further confusion for the LLM. So the quality of "pruning" might be a determining factor. This issue is also likely to arise if the example inputs used have 1) state-to-state repetitive patterns or 2) too many elements. On that note, I agree its not a fully solved problem, and can be a tricky to work for *all* algorithms. I am still of the opinion that pseudo-code in a well-defined DSL can be translated to IRSA instruction automatically for many (if not most) commonly used algorithms. Care will have to be taken when the automated translation of the algorithm induces excessive number of state variables.
> > > >
> > > > > As I said, automatically generated IRSA prompts would facilitate achieving consistency, repeatability, and reproducibility of the IRSA prompting, so we would not have the variability that impacts prompts overall and CoT prompts too.
> > > >
> > > > I agree. If provided, it will definitely make the paper stronger.

---

> > > ### Author Response · Authors · 2023-08-18
> > > **Thanks for reviewer AAHn's comments and insightful discussion (1)**
> > >
> > > We really appreciate the great comments from reviewer AAHn, as well as the comments on the significance of our work of being `of high interest to the community` and `this insight will likely intrigue researchers using LLMs at large`. We are glad that our clarifications further improved your appreciation of the paper and that you are leaning toward raising the grade. We try to address all remaining concerns and questions from the reviewer as follows.
> > >
> > >
> > >
> > > (1-a) **Precise execution**. Thanks for clarifying your argument. Re precise execution, with LLMs there are no guarantees (even GPT4 still “hallucinates” and our interesting figure in Appendix A.3.2 is most like a demonstration for this, and isn't used against our main contributions), and our main point is that the execution can actually be much closer to “precise,” **than previously thought**, though never as accurate as in LLM-in-the-loop (or LLM as translators, as we put it in the appendix). Moreover, under a circumstance with the `higher presence of recurrent patterns in its state-transition sequence`, the potential negative effect of local history patterns is very likely to be reduced by the fragmented and skip attention prompts (by removing or skipping some repetitive patterns/states in the context). In summary, we believe the combination of all ingredients in our proposed IRSA prompts will contribute to a more precise execution and better control of LLMs. We hope this will clarify some potential confusion here and make sure we're on the same page.
> > >
> > >
> > > (1-b) **Use case and motivation of why decoupling translation and execution**. Re motivation, part of our point is to inform the communities who feel that an indication of an LLM’s higher “cognitive abilities” may be its increased ability to (appropriately) execute algorithms such as constraint satisfaction, or search, graph traversal, or even just keep track of multiple alternatives. Because token stream of the LLM can be used as memory, and it can follow clear instructions well, these abilities, as long as we allow CoT, are already there. So, part of why this paper should be seen by those communities is to help them decide what and how to test. Note that mixing algorithmic and common-sense/associative reasoning can be quite subtle. If the LLM can transform the problem into a form executable by an algorithm, and can execute it, but also make an outside API call to execute it, then which parts of the algorithm will just be explained to the LLM and which will be executed exactly by that API will depend on the developer, and the application may require back-and-forth between LLM’s processing and API calls. E.g., if we want an LLM to create a few initial guesses, and then reason through them to find the best answer, this may be easier to do in an LLM-only world. On the other hand, e.g., if a landscape architect wanted to come up with a prompt that would cause LLM to iterate over possible combination of plants for a garden given some constraints, running a CSP outside of LLM may be more of a headache than justified, esp. given that the architect will select/massage the result any way. (Someone said that English is the hottest new programming language, but perhaps it will be by using it to demonstrate execution paths, possibly by people without a CS degree, rather than by translating language into code). And given the success of fragmented prompting, we expect that people will come up with some pretty creative solutions that mix algorithmic and common sense reasoning.
> > >
> > > Given the above thoughts, there are plenty of exciting use cases and we've discussed a few of them in the section of `Possible consequences` in A.3.1. The reviewer AAHn mentioned an LLM-in-the-loop system that is closely related to the section `Hybrid models - LLMs as translators`. Another potential scenario could be in environments where traditional programming languages aren't feasible or where non-specialists require a more naturalistic interface for algorithm execution. For instance, a user might need to modify or execute a procedure in real-time through natural language, without knowing the specific code. Such IRSA-enabled LLMs can bridge this gap, providing both flexibility and precision. Following the reviewer's suggestions, we will incorporate more concrete examples in the earlier part of the paper to better ground the reader's motivation.

---

> > > > ### Author Response · Authors · 2023-08-18
> > > > **Thanks for reviewer AAHn's comments and insightful discussion (2)**
> > > >
> > > > (1-c\) **Benchmark test and select the right algorithm**. Regarding benchmarks, what we saw was that each task is really a single algorithm (e.g. CSP for Logical Deduction), and the CoTs are written separately for each task (so our LD prompt is an instance of CoT solution for this task). But of course, nothing prevents the prompting to list a few possible algorithms and ask LLM to try them all, then verify which tends to solve the problem, and fragmented prompting will be of value there, too. (This is beyond the scope of the paper). Although algorithm discovery is not the topic of the paper, and besides, many (if not most) problems rely on a small subset of basic algorithms, we do believe combining all three steps mentioned by the reviewer could create enormous opportunities for more intelligent LLMs as a very promising direction to explore in the future.
> > > >
> > > >
> > > > (1-d) **Presentation, baselines, and relationship with CoT prompting**. First, we agree with the reviewer and care deeply about the importance of the paper's readability. As mentioned in the global response (2) and (1-c\) response to Reviewer tn6w, we've proposed a solid revision plan to significantly improve the presentation as well as make better use of the (additional) space to accommodate the full-page prompts. Second, we understand the importance of comprehensive baselines to contextualize our results. As mentioned in (1-b) response to reviewer tn6w, in the revised manuscript, we will include more results with GPT-4 on LCS problems in the main text for a more direct comparison with one of the most closed and competitive CoT baselines. We hope such a comparison with one of the strongest models, GPT-4 could shed light on the unique advantages of our IRSA approach over others. Last but not least, please refer to our follow-up response to reviewer n7ti, where we spend the whole response clarifying the definition of IRSA and its relationship with CoT.
> > > >
> > > >
> > > > Finally, thank you so much for addressing tn6w ‘s concerns: The topic of the paper is prompting to execute a given (known) algorithm, not automating its discovery (automated CoT prompting is a topic that is just starting to be explored by the community). We're also hopeful that our responses may prompt a reconsideration of the paper's rating. Your continued feedback is crucial to us.

---

### Official Review · Reviewer_2ucu · 2023-07-07

**Soundness:** 4 excellent
**Presentation:** 3 good
**Contribution:** 3 good
**Rating:** 7
**Confidence:** 4

**Summary:**

The paper proposed several novel prompting methods that could trigger GPT-3 to perform iterative behavior for executing algorithms with loops. The main technique the paper presented, IRSA, is to use highly structured prompts that contains information about unrolled execution trace, program states, and a detailed explanation of the motivation of a specific action, thus bringing strict attention controls to LLMs and help them reason about the procedures to get the solution.

Based on the proposed IRSA method, the authors also introduced two alternative prompting methods: Fragmented prompting and Skip attention. The fragmented prompting technique strips out some of the iterations in the full execution trace, thus enabling the prompt to contain more diverse scenarios under a limited prompt length. The skip attention technique explicitly marks program state information with a special token and puts emphasis on the original prompt that serves as a demonstration to the LLM and the last execution state where the LLM could continue its execution from, thus also bringing LLM server side and client side optimization opportunities. Meanwhile, the authors also briefly discussed the automatic generation of the proposed prompts using  LLMs.

The proposed prompting methods were evaluated on various tasks whose solutions involved loops. It is shown by the evaluations that the proposed methods could achieve state-of-the-art results on multiple tasks.

**Strengths:**


1. The paper showed great originality and insights in that the authors identified the failing reason of the LLM on tasks involving iterations, considered related Turing machine concepts, and designed several novel prompting methods incorporating highly structured information about unrolled execution trace, program states, and a detailed explanation of the motivation of a specific action, bringing strict attention controls to LLMs and help them reasoning about the procedures to get the solution.

2. The proposed prompting methods showed state-of-the-art results on multiple loop-involving tasks.

3. The proposed Fragmented prompting method can be a promising technique that could encode diverse scenarios while keeping the prompt relatively short. This can be helpful for working with LLM APIs.

4. The proposed Skip attention prompting method can also be promising in that it emphasizes the concept of program state in the context of using LLMs as general Turing machines. This technique could also help reduce prompt length, and with adequate supporting modifications and implementations on LLMs, this can be a solid base for future works.

**Weaknesses:**

The overall presentation of the work can be relatively hard to comprehend for the readers. Especially for the presentation of the proposed Skip attention, it could be better if the authors provided a figure that briefly demonstrates the idea of the server and client-side implementations of the method.

**Questions:**

1. From the presentations of the work, it seems like the potential of the proposed Skip attention method was not fully demonstrated due to the prompt length limitations of the LLM APIs. Can the authors further evaluate the method on other LLMs (with online APIs or a local deployment) that accept longer prompts? Is it possible to adapt some encoding scheme that further reduces the prompt length, thus enabling the method to be applied to more tasks?

2. Could you provide some quantitative results of the experiments on prompting to compile a program? For example, the success rate of triggering GPT3 to execute iterative algorithms on a certain task such as LCS.

---

> ### Author Rebuttal · Authors · 2023-08-10
>
> We thank reviewer 2ucu for acknowledging that our work showed great originality and insights, our prompt methods showed state-of-the-art results on multiple tasks, and the proposed Skip attention and Fragmented prompting are promising techniques as a solid base for future work. Below we address all the concerns and questions from the reviewer:
>
> (a) **Presentation and explaining Skip attention**. We address the flow (see global response 2) and demonstrate more clearly the ideas behind Skip attention, see the uploaded figure to demonstrate the implementations of skip attention.
>
> (b) **Prompt length limitation and longer context length**. Regarding the prompt length limitations, demonstrating skip attention on bigger tasks with prompt length restrictions could be explored further by utilizing shorter or more concise syntax within the IRSA prompts. In fact, skip attention allows prompt designers to address some of the length limitations that come from unrolling potentially long iterative algorithms, particularly on bigger tasks, while simultaneously avoiding confusion from accidentally generated patterns. Although the strategy should increase the number of tokens that can be generated (by effectively removing some generated text to make room for newly generated text), overall token limitations for how much the model is allowed to generate will always create a bound on the problem. That said new models, unavailable at the submission time, indeed have up to 8 times larger token limit, which will allow us to explore LCS for longer sequences in the final paper. (LCS has space requirements proportional to the product of the sequence lengths, and IRSA needs to see the whole state.) Regarding the encoding of state, that may be possible perhaps, with run-length encoding of the DP matrix, or something clever like that. But on the other hand, new models and architectures (with sparse attention, for instance) are extending the token limit anyhow.
>
> (c) **Prompting to compile a program**. In terms of providing examples of triggering execution by prompting to compile, we used prompt A.2 to compile an execution path in Prompt A.3 and used both to induce IRSA on LCS problems, the results being shown in Table 2. We can make this more clear as an example of using GPT-3 as an interpreter/compiler. We did not systematically study the interpretation of, for example, random programs, which could be a good topic for future work.

---

### Official Review · Reviewer_HgMc · 2023-07-08

**Soundness:** 3 good
**Presentation:** 2 fair
**Contribution:** 2 fair
**Rating:** 4
**Confidence:** 3

**Summary:**

The paper explores the use of regimented self-attention (IRSA) to prompt GPT-3 to perform iterative behaviors necessary for executing programs involving loops. The authors investigate three approaches to trigger the execution and description of iterations. The results suggest that IRSA leads to larger accuracy gains than using the more powerful GPT-4 for dynamic program execution. The authors highlight the potential applications of IRSA in education and discuss the implications for evaluating large language models (LLMs). While LLMs have limitations in complex reasoning tasks, prompt design plays a crucial role in their performance.

**Strengths:**

+ Interesting problem and approach
+ Providing examples of prompts

**Weaknesses:**

- Presentation
- Concern about reliability
- Concern about "Turning machine" claims


**Questions:**

There are a couple of points that could benefit from further clarification and discussion:

1. The paper mentions (Line 71-73): "it is easy to mislead with a prompt with accidental alphabetical or numerical ordering, or some undetectable semantic bias," and "slight changes in prompts can yield dramatically different responses." However, the authors do not provide any specific syntax/structure for the IRSA query prompting scheme using the CoT paradigm. It is unclear whether changing a certain text ordering in these prompts will produce the same performance. In other words, this raises concerns about the reliability and consistency of LLMs in generating accurate outputs. Can you elaborate on the robustness of IRSA to different prompt variations and its sensitivity to prompt design choices?

2. Table 1: What does guessing mean for the longest substring, logical deduction?

3. The experimental results show significant performance improvements when using IRSA on logical deduction puzzles. However, it would be valuable to understand the generalizability of these findings. How does IRSA perform on other complex reasoning tasks beyond logical puzzles?

4. The paper claims that IRSA outperforms GPT-4 on dynamic program execution but lacks a thorough comparison or analysis of GPT -4's performance in the main paper. Can you provide more insights into the limitations of GPT-4 and why it fails to consistently execute code without IRSA prompting?

5. Line 268-269: Is there proof of this? It seems a far-fetched & ambitious statement that LLMs are similar or equivalent to Turing machines. Moreover, the authors mention "... becoming a Turing machine" in the title (that's a huge claim), but there are only two other places in the paper that mention "Turing machine" in the sentence. The authors should add more details to justify this more concretely.

6. Github URL seems to be not anonymous

7. The names of the methods "Iteration by Regimenting Self Attention (IRSA)," and "Skip attention" are misleading. It makes the reader feel that it is related to the internal attention mechanism of a GPT, but actually it deals with query/prompt engineering.

8. The paper seems to be written in a hurry, so there are a couple of typos and some discontinuity while reading.

**Limitations:**

* The authors make huge claims, but do not discuss the limitations of their work.

---

> ### Author Rebuttal · Authors · 2023-08-10
>
> We thank the reviewer for acknowledging that our problem and approach are interesting and the potential applications of IRSA in education. Below we address all the reviewer's concerns and questions.
>
> (a) **Reliability**. We are using much stricter attention control than usual CoT prompts, however, even in our case there is a certain degree of sensitivity to formatting. There is an example in Table 1 with bubble sort where two different structures for prompts, while both being much better than the baseline, yield different accuracy on the same problem (100% vs 74%)
>
> (b) **Guessing baseline**. The motivation for guessing strategies is explained in lines 174-177. For the longest substring, with a problem library of fixed length strings, there is some most common length of the longest substrings among the problems, and guessing is described as guessing the most frequent length for every problem. For logical deduction, the dataset was balanced, with 5 potential answers for every problem, and without any bias in the dataset, guessing is correct 1 out of 5 times. The guessing strategy that exploits the most basic imbalances in the data made sense to us more than just guessing uniformly among possible values. Since almost any ML algorithm would learn those biases quickly, the difficulty of the task is not well represented by uniform guessing.
>
> (c\) **Other complex reasoning tasks**. We were using the logical deduction example to address the point that many tasks in the benchmarks are addressing problems for which there are already algorithms. If you have a natural language representation of a reasoning task, we have shown you can get the LLM to translate the problem into a canonical form using CoT reasoning. If algorithms are known and CoT reasoning is allowed, we can get the LLMs to first translate the problem into a canonical form where the algorithm can be applied, and then it can even apply the algorithm itself. If other complex reasoning tasks are of this sort (there is an algorithm that can be applied to solve it and the problem is in a natural language form such that it can be translated to a form the algorithm can work on), then we posit that it would be possible with sophisticated enough CoT prompting using techniques described in our paper to solve such tasks. As we discussed in the general response, this is important for the community to understand. CoT designs can be very sophisticated (as IRSA is) and so comparing LLMs is highly dependent on prompting. We used Logical Deductions to show how large the difference can be, begging the question: Was that the best one can do? Do we ever know what the best is?
>
> (d) **Discussing the limitations of GPT-4**. Regarding the limitations of GPT-4, we discuss this in the Appendix (lines 599-615, section A3.3). Do you suggest it should be moved to the main paper? In the case of the LCS problem, GPT-4 and GPT-3 do not need to be prompted with code, as they can generate it. Thus one might imagine that a CoT that asks for the code to be written and then executed step by step should perform similarly to our prompts. Yet, GPT-4, while better than GPT-3 under such prompts, still only gets 63% accuracy compared to 93% using IRSA. To illustrate why, we showed how different problems get processed differently using the same prompt. Prompts A.11 and A.12 both ask GPT-4 to recall a dynamic programming algorithm in Python and write down its execution with intermediate steps to solve the length of the longest common subsequence for two sequences. Their only difference is the strings used as input for the problem. Although GPT-4 successfully recalls the algorithm in Python for the problem and makes some attempts to execute it, it is not consistent in how it shows the execution with intermediate steps. In the case of prompt A.11, GPT-4 shows the initialization of a table, then immediately displays what the completed table looks like after iteration, and gives its answer. However, in prompt A.12, GPT-4 initializes the table and then displays a couple of steps, before jumping again to the end table and giving its answer. This inconsistent processing means that some answers may be (impressively) correct and thorough, while in other cases the LLM will just skip steps or start hallucinating. A more regimented prompt showing key fragments, or entire execution (as in our examples) seems to be needed to get consistent results.
>
> (e) **Claims on Turing machine**. Regarding Turing machines, see the general response.
>
> (f) **Use of "attention" in names**. Regarding the use of 'attention' in the names, all of this is possible because we can direct the attention of the model. Basic IRSA does it with prompt design, and skip attention literally prevents LLM from seeing some of its previously generated tokens because we want it to look only at the last full state. This is doable on either the server side or the client side (see the uploaded illustration figure), and then calls the server again to transform the new state; The client-side solution of skip attention keeps reprocessing the prompt, while the server could simply keep the states associated with it and just block attention to the text generated before the latest <state> ... </state)> structure, thus saving on token quota and computation and the implementation of it would be similar to how stop words are implemented by OpenAI API).
>
> (g) **Discussing the limitations**. Regarding the limitations, we discuss them in a few places. One, we show that there is a variation in performance on the same task with different IRSA-styled prompts (though they are both much better than the baseline, Table 1). Second, the appendix has a section on limitations, including a worrying problem with LLM's cryptic dependence on ordering of statements (Fig. A.1). However, with some experience, summarized in Section A.3.2, we found it possible to find prompts that lead to significantly superior results compared to SOTA.

---

> > ### Author Response · Authors · 2023-08-19
> > **Thanks for the review and looking forward to your feedabck to our rebuttal**
> >
> > Dear Reviewer HgMc,
> >
> > Thank you for the time and expertise you've shared through your feedback on our paper. We've taken your comments to heart and have made appropriate revisions in response.
> >
> > As the author-reviewer discussion phase nears its conclusion, we wish to bring to your attention the constructive discussions we've had with the other reviewers. Through these engagements, we've made substantial progress in refining our paper and emphasizing its core contributions. Encouragingly, a consensus among the majority of the reviewers is seemingly emerging, recognizing the significant contributions our IRSA prompting offers to the broader community.
> >
> > We understand the many commitments reviewers such as yourself have, and we truly appreciate the time and effort you've already dedicated. We genuinely hope that our revisions and responses resonate with your observations, and we are eager to incorporate any further suggestions you may have.
> >
> > Thank you once again for being a pivotal part of this academic journey. Your continued engagement and feedback are invaluable to us.
> >
> > Best,
> > The authors of Paper 8965

---

### Official Review · Reviewer_n7ti · 2023-07-08

**Soundness:** 3 good
**Presentation:** 2 fair
**Contribution:** 3 good
**Rating:** 6
**Confidence:** 3

**Summary:**

This work introduces Iterations by Regimenting Self-Attention (IRSA) which is a set of LLM prompting techniques for producing repetitive, algorithm-like behavior that can be useful for a range of tasks, such as carrying out a sorting algorithm or solving a logic puzzle. There are 3 techniques discussed: 1) "Basic IRSA" which is a chain of thought prompt that looks a bit like an execution trace of some natural-language-like pseudocode - theres a lot of repetitive structure, the current state is verbosely repeated after each step, and changes to the state are explicitly describe before the happen. 2) "Fragments" which is the idea that instead of prompting with a full trace of an algorithm you can just prompt with random unordered individual steps of the algorithm to prepare the model for executing a random step. 3) "Skip Attention" where only the most recently produced state is attended to (plus the original, fragment-based prompt), since changes to the state should be independent of the history of states – this cuts down on computation and helps the LLM not get confused by patterns in its recent output.

**Strengths:**

- Skip attention and fragments (which pair well together) are great ideas, and are original as far as I know – other reviewers can correct me if I'm wrong. In many algorithms (and in fact, in the execution of interpreted code in general) only the current state matters as opposed to the history of how the state has changed. Only showing the most recent state to the LLM makes a lot of sense. It saves on computation cost and makes long running algorithms feasible, since there's no need to attend over the whole history of generations (which would become a huge problem as an algorithm runs for dozens or hundreds of steps). As the authors point out, this Markovian setup of not looking at the history of states also means that the LLM won't get confused by patterns in its recent history.
  - The "fragments" approach is a clever way to get the LLM used to this idea of seeing somewhat random states and needing to do a single algorithmic step for each one.
  - Skip attention makes so much sense, I'm surprised past work like "Show Your Work" (Nye et al 2021) didn't take an approach like this, since I imagine it would work fine with executing interpreted Python programs (where the state is the set of local variables/values along with the current line number in the program, and the LLM just has to output a next set of local variables and next line number).
- More generally, getting LLMs to do things that look more like rigid computation can be difficult and I think that this is a paper with a pretty good evaluation of a particular approach to this problem, and would be useful for the NeurIPS community to see.
- The evaluation is reasonable and shows unsurprisingly that the skip attention method can work great and generalize to very long sequences (eg bubble sort with 25 steps).

**Weaknesses:**

- The descriptions of what IRSA is were quite difficult for me to understand. The first line of section 2, the section describing IRSA, is "Prompt 1, as well as the prompts 2, A.4, A.5, and A.6 in the Appendix, illustrate the basic IRSA." (line 66). Written as is this feels a bit overwhelming as it suggests that I need to look at 5 different prompts (including 3 in the appendix) and try to look for the common features among them to figure out the method. Also, in reality many of these references (2, A.4, A.5, A.6) are actually going to show up later on in places where they're discussed so it's okay if I don't look in detail at them now, but since I haven't been told that I feel some need to dig them all up before continuing.
    - A flow for section 2 that would be much more understandable to me (and I believe others) would be the following. This is just one suggested way of doing it and I think there are many valid ways that would be widely understandable (you dont need to do the below), but the current flow is difficult to understand:
        - Give a brief but concise description of the key feature of IRSA (similar to lines 71-73 right now) so we're primed with looking for that *before* we're told about any prompts to look at. I might even suggest that instead of putting the CoT comparison at the end (lines 78-82), it might flow nicer to actually frame it *in terms of CoT / as an extension building on CoT* since that is a closely related framework many readers know about. In general after reading the paper I actually still find I have trouble precisely articulating what makes something count as "basic IRSA", so presenting it from the start in terms of its relation to CoT might be helpful.
        - Tell us to look at Prompt 1, and briefly walk us through what we're looking at / why this is IRSA.
        - Mention that the precise keywords/format of Prompt 1 ("EXECUTION", "Prep", "EndPrep" "Iteration", the indentations, "State:") are not important (IRSA is not a set of specific keywords to use) and point to Prompt 2 as an example of something that looks different on the surface level but is still IRSA.
        - At this point, you might parenthetically refer to the 3 appendix prompts as additional examples used in the evaluation that the reader can look to if they want more.
    - Again, to be totally clear, I'm not prescribing this format, I just find the current flow difficult to understand so I took a stab at restructuring it, but there are many other ways of doing so that would also flow well.
- I'd like to see some discussion of how this relates to the paper "Show Your Work: Scratchpads for Intermediate Computation with Language Models" (Nye et al 2021) which is currently just referenced by the paper in the list of CoT related works without specific discussion. In that work, the authors showed that while LLMs are bad at directly predicting the output of a Python function called on certain inputs, they could instead have the LLM repeatedly output the current state (what the variables are set to) plus the next line of the program to run. That was essentially a form of CoT with extra structure. IRSA seems somewhere in between the strict state/instruction format of Show Your Work and the free flowing reasoning of more general CoT. I think an explicit comparison to that paper (and/or any other paper that does some form of rigid CoT) is important, so it's clear how this work should be viewed in relation to others that have structured CoT.

- I'm generally coming out of this paper still somewhat unsure what precisely "basic IRSA" is (i.e., without fragmenting or skip attention), and I felt I could only gesture towards some of its important features when writing the Summary section above.
    - It feels related to CoT and I'd like to understand it in terms of that. One section comparing CoT to IRSA says "a significant distinction lies in the number of reasoning steps, which is limited and fixed in *usual* CoT applications" (emphasis is mine) (lines 78-82) but this is not always true (e.g. in the Show Your Work paper above – so does that make Show Your Work and instance of IRSA and this present paper is proposing a general framework encompassing that?).

- (minor weakness) Section 2.4 sounds interesting but is largely confined to the appendix. It doesn't really flow with the rest of the story and as far as I can tell isn't used in the evaluation alter. But I'm a bit torn because it is actually quite cool and maybe it doesn't hurt to have as just an aside (though maybe with a slightly more clear verbose explanation). I don't terribly hold this one against the paper, it just feels a little out of place.

**Overall** I think that, though it is difficult to follow the flow of this paper in places so it took me quite a while to understand, and I'm still not totally clear on what makes something "basic IRSA" or how it relates to prior work like Show Your Work, I think that in particular given the contributions of skip attention and fragments this would still be valuable work for the NeurIPS community to see. The positives outweigh the negatives in my view, but with revisions around the points mentioned above I would be more supportive.


**Questions:**

- (as discussed in Weaknesses) How does IRSA relate to the "Show Your Work" paper?
- (as discussed in Weaknesses) How, precisely, does IRSA relate to CoT?
- While full LLM prompts can be useful, using 3 full pages for three full-page LLM prompts is a lot, and you might consider abbreviating some of these prompts to keep the key bits (while leaving the full version in the appendix).
- "world" -> "word" typo in the bolded text of line 85
- "the Prompt 2 Appendix" (line 233) seems like some sort of typo, is this prompt 2 or prompt A.2 or something else?
- For Table 1 Instead of saying Prompt 1 and Prompt A.4 I would say things like "Base IRSA (Prompt 1)" or something, so that at-a-glance you can understand these results without having to remember / look up what exactly prompt 1 and prompt A.4 are. It'd also be helpful for the entries like "Longest substring" to actually say which variant of IRSA was used for this result (or to put that in the caption, or to put it as separate columns, or do anything else that just makes it easy at a glance to see which IRSA method you're talking about).


**Limitations:**

The authors address limitations adequately

---

> ### Author Rebuttal · Authors · 2023-08-10
>
> Thanks a lot for acknowledging that our contributions of skip attention and fragments this would be "valuable work for the NeurIPS community", our approaches are novel and original, and our evaluation is reasonable that the skip attention method can work great and generalize to very long sequences. Below, we address all the comments and questions from the reviewer.
>
> (a) **Writing flow**. Thanks for the thoughtful comment! Following your suggestion, we will split line 66 to indicate that Prompt 1 is enough to demonstrate basic IRSA and further prompts are just other examples of basic IRSA. We like the idea of explaining Prompt 1 and emphasizing that IRSA is not about which specific keywords or structures are used, but the consistent and repetitive use of those chosen within a single prompt. In other words, we will restructure Section 2 to flow closer to the reviewer's suggestion. Regarding our use of full prompts, our concern was that partial prompts may confuse the reader, since they may not be able to understand the basic idea without going to the appendix to see the rest of it (not to mention the confusion that may happen once the reader moves on to the fragmented prompts). With the additional page allowed for the final version, if accepted, we will make better use of the space to adjust the space for the full prompts and their explanations, following your suggestions.
>
> (b) **Typos and captions**. Thank you for your notes on readability and we will address those.
>
> (c\) **Relateness to "Show Your Work"**. See also the general response regarding the flow as well as the relationship with "Show your work...", Nye et al.; we also give examples of IRSA prompts for their, much simpler, problems. Note that we primarily focus on ways to explain to an LLM how to execute a given algorithm, rather than to learn how to execute Python; the former is an instance of programming an LLM in its "native language", the latter an instance of interpreting or compiling, which we also touched upon in the paper.
>
> However, it is also true that skip attention can be used in Nye et al or other models. We appreciate the comment on applicability beyond our work. It also appears that Nye et al could have had better results even just by regimenting their prompts and training data a bit more strictly as we show in the [prompts](https://platform.openai.com/playground/p/d5lzF0FOQG31NcyGAhE96JuZ?model=text-davinci-003) in the general response. For example, their first example in Appendix C would be better given like this:
>
> ```
> Program:
> def f(v0):
>     v0 += 0
>     v4 = 2
>     while v4 > 0:
>         v4 -= 1
>         v0 *= 2
> return v0
>
> Call:
> output = f(3)
>
> BEGIN
> We first pass the input of f(3) to the function. It is assigned to v0
> state: v0=3
> command: v0 += 0
>     What is  v0 in the state? 3. v0 is set to v0+0=3+0=3. New state is:
> state: v0=3
> command: v4 = 2
>     We set v4 to 2 and keep v0 as is. New state is:
> state: v0=3, v4=2
>
> Iteration:
> command: v4 -= 1
>     What is v4 in the state? 2. v4 is set to v4-1=2-1=1. New state is:
> state: v0=3, v4=1
> command: v0 *= 2
>     What is v0 in the state? 3. v0 is set to v0*2=3*2=6.  New state is:
> state: v0=6, v4=1
> check for iteration end. v4 is 1. And 1>0 is true, so we need more iterations.
> Iteration:
> command: v4 -= 1
>     What is v4 in the state? 1. v4 is set to v4-1=1-1=0. New state is:
> state: v0=6, v4=0
> command: v0 *= 2
>     What is v0 in the state? 6. v0 is set to v0*2=6*2=12.  New state is:
> state: v0=12, v4=0
> check for iteration end. v4 is 0. And 0>0 is false, so we end the iteration.
> Final state is:
> state: v0=12, v4=0
>
> What is v0?
> Answer: 12.
> END
> ```
> In fact, as the second playground link shows, following this with their second program prompts GPT into executing it.

---

> > ### Comment · Reviewer_n7ti · 2023-08-16
> >
> > I appreciate the authors thorough response and I'm glad to hear about the flow revisions.
> >
> > I really appreciate the extensive discussion of how this work relates to Nye et al (and the executed playground examples!), this clarifies things for me. Having some very brief mention of this relationship in the final version related work or intro would be useful context for where this work sits.
> >
> > My original review had some concerns around clarifying how IRSA relates to CoT, and what precisely makes something count as basic IRSA. I hope the authors' planned improvements to the explanations of Prompt 1 at the start of Section 2 will help readers with this. However I wanted to spend a little more time below discussing this to make sure that it is clear in the final version since the authors didn't explicitly bring up the CoT relation in the rebuttal (though I appreciate the time they spent on the particular Nye et al example):
> > 1. The paper's current wording in certain places feels like it's claiming that IRSA is *distinct* from CoT prompting, when really it seems to be a particularly effectively *variant* within the broader paradigm of CoT prompting. I'm assuming the authors intend the latter interpretation of IRSA's relation to CoT because in the general response to all reviewers, the authors say "Defined as anything with step-by-step instructions, CoT prompting includes IRSA solutions". In the paper, lines 78-82 and 149-158 seem to put CoT at odds with IRSA ("However, a significant distinction..." and "Although similar to CoT prompting, there are notable differences..."), and very slight rewording could clarify that actually the features don't create a distinction between CoT and IRSA, but rather that these are the specific features that make an instance of CoT considered IRSA.
> > 2. Assuming the authors frame IRSA as a variant of CoT, then what makes something IRSA is, in my understanding:
> >     - A. The prompt shows all state changes and explain each change before it occurs, using a rigid repetitive explanation (71-72)
> >     - B. The prompt contains condition for declaring end of execution, so it can run for an unspecified number of iterations (80-82)
> >     - Please correct me if I'm wrong or there's more I'm missing
> >     - Assuming A and B are what define IRSA and are *both* things that make a variant of CoT considered IRSA, I think some revisions could clarify this around the two places where IRSA is detailed and CoT comes up:
> >         - In 78-82, right now CoT is brought up right between A (71-72) and B (80-82) even though both A and B are about how IRSA differs from CoT, so just putting the CoT mention either before or after would be fine.
> >         - In 149-158, the same concern applies where CoT appears between A and B.
> >         - In 149-158 the version of A is "Prompting with highly structured single execution path examples" which is a little vaguer than the 71-72 version but I think that's okay since it's consistent with the language used in the abstract and intro.
> >         - In 149-158 After the mention of CoT, B is listed ("iterative reasoning that is repeated until the stop condition is reached") but then yet a third thing, call it C, is listed: "Furthermore, the execution path example for each task is deliberately chosen to be out-of-distribution". This leaves me a little confused on if C is a third feature of what makes something IRSA that wasn't mentioned in the 78-82 section (in which case perhaps it should be)? Or if it's just an elaboration on how B allows for more flexible examples (in which case maybe it shouldn't be listed as two separate distinctions but rather as B + a note on the flexibility that B allows for)?
> >     - My broader intention with all of this that I came away from this paper with trouble listing the essential features of IRSA and how it relates to CoT (even more broadly than with how it relates to Nye et al, but I very much appreciate the authors extensive discussion of the relation to that!) and so in this follow up I'm trying to nail down the points that really confused me and give some thoughts/suggestions on how they could be made more clear. I think there are some super exciting ideas in this paper and I want to make sure the groundwork laid by the "basic IRSA" section is clear. Of course, let me know if any of these suggestions are coming from my own misunderstanding of the authors' intentions.
> >
> > Thank you again for the care you've taken in the rebuttal/revisions

---

> > > ### Comment · Reviewer_tn6w · 2023-08-17
> > > **Thank you reviewer n7ti regarding your comments on CoT and IRSA**
> > >
> > > I think reviewer n7ti makes important points about defining IRSA and its relationship with CoT.
> > >
> > > Pursuing these clarifications would improve the paper.

---

> > > ### Author Response · Authors · 2023-08-18
> > > **Thanks for reviewer n7ti's comments and insightful discussion**
> > >
> > > We thank a lot for reviewer n7ti's always thoughtful and insightful comments and discussions. We are happy to see that our discussion on Nye et al clarifies the reviewer's question. We surely will add them and further enrich the related work section. We now try to address the remaining concern on **clarifying the definition of basic IRSA and its relationship with CoT**.
> > >
> > > First, we thank the reviewer for pointing out the text lines to help clarify the definition of basic IRSA for a better understanding of its relationship with CoT. As pointed out by the reviewer, points A and B do indeed define the basic IRSA as a highly regimented prompt that can be seen as part of the CoT family (and more broadly the art of prompting, as CoT can refer to almost anything that includes hints on how to perform a task).
> > >
> > > Second, with C, we point out the practical deviation from the usual application of CoT in in-context learning. CoT was initially demonstrated and usually applied, by making prompts that include a few instances of a task from the dataset in a step-by-step manner. However, we see that when the prompt is regimented with A and B, then it often works on out-of-domain problems (e.g., sorting letters, or objects by size instead of numbers, even though the prompt was given for numbers). In particular, in Bubble Sort experiments, our prompt demonstrated the execution not on an example from a dataset, but on a shorter sequence. Because the prompt focuses on the logic of the algorithm execution, like programs, it can generalize to longer sequences (which is related, but not quite the same, as shown in point B, though the reason why this generalization is possible is the same: regimented prompting that shows an LLM what to do under different conditions, thus “programming” it to execute an algorithm).
> > >
> > > Finally, when we found this out, we decided to attempt fragmented prompting, which further deviates from typical CoT applications: Instead of working out full examples in prompts, it is enough to show fragments of several, achieving better coverage with much shorter prompts. And then, fragmented prompts pointed to Skip-attention or skip-to-state (which can be done with basic IRSA, too, both to save on tokens and to avoid unnecessary long text that may or may not confuse LLM’s attention mechanism; skip-to-state draws it only to the latest state).
> > >
> > > At any rate, we're very grateful for the fruitful and engaging discussion with reviewer n7ti and for pointing out that `there are some super exciting ideas in this paper`. Your suggestions are welcome and will improve the final version of the paper. We're also hopeful that our responses may prompt a reconsideration of the paper's rating. If there are any other questions, please let us know.

---

> > > > ### Comment · Reviewer_n7ti · 2023-08-21
> > > >
> > > > Thank you very much for this response
> > > >
> > > > > As pointed out by the reviewer, points A and B do indeed define the basic IRSA as a highly regimented prompt that can be seen as part of the CoT family
> > > >
> > > > Thank you for clarifying this. I trust the clarification of this in the paper (through what I suggest or something else that makes the CoT relationship and definition of basic IRSA clearer) will be implemented in revisions.
> > > >
> > > > > However, we see that when the prompt is regimented with A and B, then it often works on out-of-domain problems
> > > >
> > > > Got it. I think that in that case, minor rewording could clear up that A + B constitutes IRSA, and C is a benefit of using IRSA.
> > > >
> > > > > Finally, when we found this out, we decided to attempt fragmented prompting, which further deviates from typical CoT applications
> > > >
> > > > Yes, agreed that skip attention and fragmenting are interesting further extensions beyond CoT! My discussion has largely been around basic IRSA since most of my confusion lay in that, but I appreciate that there is even more to the paper.
> > > >
> > > > I've read over the discussions with other reviewers as well and appreciate the points they've raised. I also agree with tn6w's mention of how significant enough changes have been suggested throughout the discussion in many threads that I'm left a little unsure how the final paper will look. I remain in support of acceptance though I don't feel confident enough in the final result to raise above my current Weak Accept. Thank you for the discussions.

---

### Author Rebuttal · Authors · 2023-08-10

We thank the reviewers for the great comments and constructive suggestions. We are encouraged that reviewers agree that the paper provides "Interesting problem and approach,” that fragmented prompting and skip attention, "which pair well together", are original and can have applications beyond prompting ("Skip attention…, I'm surprised past work like Nye et al 2021 didn't take an approach like this"), and that "methods showed state-of-the-art results on multiple loop-involving tasks."

We have addressed all comments and questions from reviewers with thorough clarifications and discussions via separate responses. There are three common comments we'd like to address.
1. **Motivation/precise language on Turing machine**. TMs are a theoretical construct, involving infinite memory, able to execute arbitrary algorithms. Practical computers execute only algorithms that fit their memory. That an LLM can be, other than for finite memory, Turing-complete is not a huge claim per se. Simple 2-tag systems are Turing-complete and GPT can emulate them (until it runs out of tokens). Schuurmans [1] has shown that an LLM with access to infinite memory can emulate TM U_{15,2}. The spirit of our title is that GPT is not only theoretically able to execute arbitrary algorithms with loops, but that there is a way to instruct these models to do so with natural language. Program execution has been of interest to the LLM communities, but the results were underwhelming, and we show that they can be much better, with two caveats. One, like computers, LLMs have limited memory (tokens), which can be ameliorated with skip attention. Two, there are no theoretical guarantees. But, empirically, given our results, it is possible to prompt an LLM into a high level of consistency in execution. To the community of researchers studying the ability of LLMs to execute algorithms, this is immediately important. And, several benchmark reasoning tasks target word problems that are solvable algorithmically, as long as the wording can be translated into a machine-readable form. The community studying CoT in that context should understand that LLMs can perform both the translation and the execution (as illustrated in Logical Deductions). Defined as anything with step-by-step instructions, CoT prompting includes IRSA solutions, which are approximately equivalent to just using an LLM to translate the word problem into algorithm input and then running it on a computer (Section A.3.2, LLMs as translators).
2. **Presentation flow**. While some reviewers consider the paper "well presented" with "the prompt examples very useful in understanding the idea", others find it difficult to parse in places. The flow will be easily but significantly improved using the reviewers’ suggestions. Prompt 1 is indeed sufficient to explain the basic IRSA, and the other examples can be listed after describing its features: It covers the full execution path;  The keywords can be chosen arbitrarily, but should be used consistently (instead of "set" we could have used "change", and instead of "State: can be replaced by  <state>  and so on); Importantly, the basic IRSA prompt shows conditions for starting a new iteration as well as when all the iterations are finished. We also upload an illustrative figure on how it works on the server/client.
3. **Relationship with Nye [2]**, which also involves execution paths in their scratchpad. Nye [2] target much simpler programs as they argue that "GPT-3 struggles to perform addition on numbers with greater than 3 digits" and that LLMs "struggle to predict the result of executing Python." They train/tune models with execution paths. Also, with low success, they attempted in-context learning (their Appendix C). We show that GPT-3 can produce execution paths for a *fixed algorithm* on *new* inputs given just a prompt that describes how that algorithm should be executed (no tuning). In other words, we described a technique for programming LLMs, not for teaching them how to interpret (Prompt 1 is a Bubble Sort “program for GPT”). However, our Compiler/interpreter prompt A.2 does show an example of how we can prompt GPT into creating an execution path for a new program, and we used it to get an execution path for LCS that we then used in our experiments as an LCS “program for GPT”. The difference between our approach to interpret and Nye [2] is that we described the syntax step by step (from assignment, to memory retrieval, to basic loop) rather than giving a few examples of programs/executions and asking LLM to infer the commonalities. In addition, the Nye [2] Appendix C falls for a few pitfalls of non-linear exposition we described in Section A.3.2, e.g. by not explaining the action before it is done and clearly describing iteration decisions at the right time. An example of the IRSA-style execution path for the first example in Nye [2] Appendix C is here: https://platform.openai.com/playground/p/d5lzF0FOQG31NcyGAhE96JuZ?model=text-davinci-003. With this prompt GPT can compute the output of their first f() function for different inputs. Furthermore, when the Python program precedes that execution path in the prompt, then GPT may execute a new program, too (although a better compiler/interpreter is our Prompt A.2). E.g, 1st program in Nye Appendix C and the execution above are used to interpret their 2nd program here: https://platform.openai.com/playground/p/yGFDaSaVtZWdDOZOP37sy7Dw?model=text-davinci-003
The reviewers correctly observe that skip attention can be used in Nye [2] models, and more generally in training new architectures.

**References**:

[1] Schuurmans, Dale. "Memory augmented large language models are computationally universal." arXiv preprint arXiv:2301.04589 (2023).

[2] Nye, Maxwell, Anders Johan Andreassen, Guy Gur-Ari, Henryk Michalewski, Jacob Austin, David Bieber, David Dohan et al. "Show your work: Scratchpads for intermediate computation with language models." arXiv preprint arXiv:2112.00114 (2021).

---

> ### Author Response · Authors · 2023-08-10
> **Thanks for the reviewers' acknowledgment and encouragement**
>
> We really thank the reviewers for their great comments and encouragement of our work for acknowledging that:
> 1. our proposed techniques, especially Skip attention and fragmented prompting, "pair well together," are both "great ideas" and "original", "Skip attention makes so much sense" and fragmented approach is "clever" (R-n7ti), and that our method is "somewhat novel" in enabling LLMs to mimic the execution of iterative programs (R-tn6w), offering a "novel and original direction" (R-AAHn) and showing "great originality and insights" (R-2ucu);
> 2. our approach is efficient, "Only showing the most recent state to the LLM makes a lot of sense", saving on "computation cost and making long running algorithms feasible" (Reviewer n7ti), with the "skip attention method that can work great and generalize to very long sequences" (Reviewer n7ti), and leading to "larger accuracy gains" (Reviewer HgMc) and "state-of-the-art results on multiple loop-involving tasks" (Reviewer 2ucu);
> 3. the breadth and clarity of our paper are commendable, with comments praising the quality and presentation of the paper, including figures and tables (R-AAHn), and noting the paper as being "clearly written and well presented" (R-AAHn) with "well written" examples (R-tn6w).
>
> We are also thrilled to find that our work is seen as having significant potential applications and implications for future research, especially in the fields of education and rigorous computational tasks (R-HgMc, R-2ucu, R-AAHn). Moreover, our work is considered as being "valuable work for the NeurIPS community", "a solid base for future works" and "of interest to the research community at large" (R-n7ti, R-2ucu, R-AAHn).

---

### Decision · Program_Chairs · 2023-09-21

**Decision:**

Reject

**Comment:**

The paper introduces a new family of prompting techniques, Iteration by Regimenting Self-Attention (IRSA), for LLMs to perform iterative behaviors needed when executing algorithms with loops, e.g., bubble sort. More concretely, three prompting techniques are introduced in this family, differing in how much information is provided in the prompt and how it is structured. Evaluation is done on a variety of challenging programming tasks, and the results show the efficacy of IRSA prompting. The reviewers acknowledged that the paper introduces interesting prompting techniques and appreciated the extensive experimental evaluation. However, there was some spread in the reviewers' assessment, and the reviewers raised several concerns in their initial reviews. We want to thank the authors for their hard work preparing the detailed responses and actively engaging with the reviewers during the discussion phase. These responses did help in improving the reviewers' assessment of the paper; however, the paper's ratings still stand below borderline. One of the concerns is that a significant number of changes have been suggested for updating the paper, and the revised paper would benefit from a thorough review. Based on the current ratings and discussions, unfortunately, the final decision is a rejection. Nevertheless, this is potentially impactful work, and we encourage the authors to incorporate the reviewers' feedback when preparing a future paper revision.